 # RedPajama: an Open Dataset for Training Large Language Models

**Maurice Weber[1], Daniel Y. Fu[1,2], Quentin Anthony[4,8,10], Yonatan Oren[1]**
**Shane Adams[1], Anton Alexandrov[7,], Xiaozhong Lyu[7], Huu Nguyen[5], Xiaozhe Yao[7],**
**Virginia Adams[1], Ben Athiwaratkun[1], Rahul Chalamala[1,11], Kezhen Chen[1], Max Ryabinin[1]**
**Tri Dao[1,6], Percy Liang[1,2], Christopher Ré[1,2], Irina Rish[8,9], Ce Zhang[1,3]**

[1] Together AI,   [2] Stanford University,   [3] University of Chicago
[4] EleutherAI   [5] Ontocord.ai,   [6] Princeton University,   [7] ETH Zurich
[8] Mila, Montréal, Canada   [9] Université de Montréal   [10] Ohio State University   [11] Caltech

## Abstract

Large language models are increasingly becoming a cornerstone technology in artificial intelligence, the sciences, and society as a whole, yet the optimal strategies for dataset composition and filtering remain largely elusive. Many of the top-performing models lack transparency in their dataset curation and model development processes, posing an obstacle to the development of fully open language models. In this paper, we identify three core data-related challenges that must be addressed to advance open-source language models. These include (1) transparency in model development, including the data curation process, (2) access to large quantities of high-quality data, and (3) availability of artifacts and metadata for dataset curation and analysis. To address these challenges, we release RedPajama-V1, an open reproduction of the LLaMA training dataset. In addition, we release RedPajama-V2, a massive web-only dataset consisting of raw, unfiltered text data together with quality signals and metadata. Together, the RedPajama datasets comprise over 100 trillion tokens spanning multiple domains and with their quality signals facilitate the filtering of data, aiming to inspire the development of numerous new datasets. To date, these datasets have already been used in the training of strong language models used in production, such as Snowflake Arctic, Salesforce's XGen and AI2's OLMo. To provide insight into the quality of RedPajama, we present a series of analyses and ablation studies with decoder-only language models with up to 1.6B parameters. Our findings demonstrate how quality signals for web data can be effectively leveraged to curate high-quality subsets of the dataset, underscoring the potential of RedPajama to advance the development of transparent and high-performing language models at scale.

## 1   Introduction

Pretraining data is among the most central building blocks that go into the development of modern large language models (LLMs). However, one of the core challenges this field faces is the general lack of transparency regarding the composition and curation strategy of pretraining data [8]. Indeed, with a few notable exceptions [19, 4, 2, 65], the majority of reports documenting state-of-the-art LLMs [1] provide scarce details, if any, on their pretraining datasets. Even open-weights models such as LLaMA [57, 58] provide little to no details about their training data, let alone release their datasets. Furthermore, the process of studying and building optimal data compositions, along with developing filtering rules and heuristics, is time-consuming as it necessitates running numerous ablations on

38th Conference on Neural Information Processing Systems (NeurIPS 2024) Track on Datasets and Benchmarks.

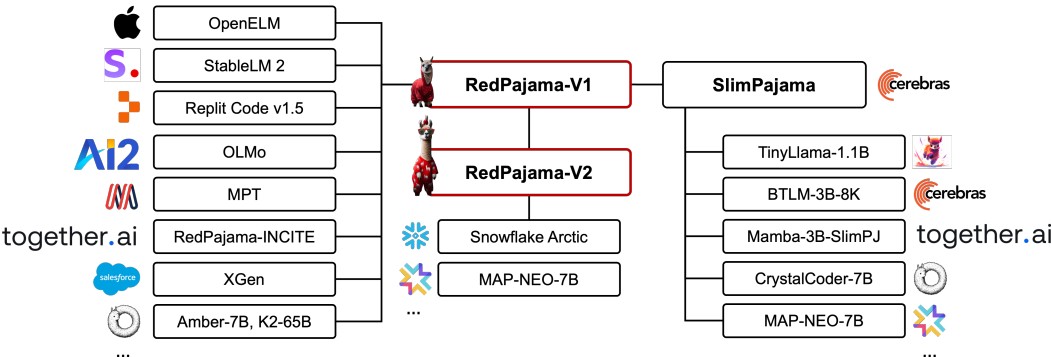

Figure 1: The ecosystem around the RedPajama datasets. RedPajama has provided pretraining data for multiple open-source LLMs, including OpenELM [36], OLMo [19], Snowflake's Arctic [54] and RedPajama-INCITE. SlimPajama is a cleaned and deduplicated version of RedPajama-V1.

different compositions of the training data. To address these challenges, and with the overarching goal of democratizing the access to and development of open-source LLMs, we have released the RedPajama datasets, which in total consist of more than 100 trillion tokens of text data. With this goal in mind, we use the following design principles to guide our approach to creating open datasets:

**Transparency.** We strive for maximal *transparency* from at least two angles. On the one hand, this entails documenting and making all aspects of data curation publicly available[1]. On the other hand, we strive for open and transparent datasets, which allows for application developers and researchers alike to better understand and design language models.

**Scale.** Large pools of accessible data are one of the core building blocks of the most powerful large language models [52, 11, 58, 1], yet are hard to come by due to the large amount of resources and expertise required to build, curate and store them. Next to transparency, we thus also strive for *scale*.

**Versatility.** We aim to provide the community with datasets and artifacts for building state-of-the-art open language models by providing a versatile resource. Rather than prescribing what constitutes a high-quality dataset, we offer a broad, general-purpose corpus of web documents. Each document is tagged with quality signals, empowering users to make informed decisions based on their specific needs and criteria.

Following these principles, we have developed and made publicly available the RedPajama datasets for LLM pretraining. RedPajama-V1 is a publicly available, fully open, best-effort reproduction of the training data described in [57], used to train the first iteration of LLaMA family of models. Together with this dataset, we have developed the RedPajama-INCITE models, which include a base, instruction-tuned, and chat models at the 3B and 7B scales. Based on the first set of learnings from these efforts, we have built the RedPajama-V2 dataset. This latter dataset takes an entirely different approach and focuses exclusively on web data. It consists of five languages, sourced from 84 Common Crawl snapshots ranging from 2014 to 2023. In addition to the raw text data, we also publish quality signals accompanying each document in a 50T token subset of the corpus with the goal of fostering research towards principled understanding of ways to filter web data. In addition, in this work, we present a series of ablation studies, showcasing the value of the quality signals for creating subsets of the raw corpus of varying quality and subsequent model performance.

In summary, in this work, we make the following contributions:

**C1** We publish the RedPajama-V1 dataset, an open reproduction of the dataset used to train LLaMA-1 [57]. We also include a detailed report on considerations that went into creating the corpus.

**C2** We report the process and evaluations on the training of the RedPajama-INCITE models, including details on how we used the Summit supercomputer and the challenges we had to address.

**C3** We publish the RedPajama-V2 dataset, the largest open pretraining dataset consisting of raw, unfiltered data scraped from the web, together with 46 measures of quality computed for each document, to enable further research in data curation.

---

[1]The code to reproduce RedPajama is available at `github.com/togethercomputer/RedPajama-Data`.

Table 1: Comparison of open pretraining Datasets along the dimensions of transparency, versatility, and scale.

| Dataset | Transparency | | Versatility | | | Scale (TB) |
|---|---|---|---|---|---|---|
| | Open Access | Open Code | Raw Data | Composite | Multilingual | |
| Refined Web [44] | ✔(subset) | ✗ | ✗ | ✗ | ✗ | 2.8 |
| FineWeb [43] | ✔ | ✔ | ✗ | ✗ | ✗ | 93.4 |
| FineWeb-EDU [43] | ✔ | ✔ | ✗ | ✗ | ✗ | 8.8 |
| C4 [46] | ✔ | ✔ | ✗ | ✗ | ✗ | 0.3 |
| mC4 [63] | ✔ | ✔ | ✗ | ✗ | ✔ | 9.7 |
| DCLM baseline [30] | ✔ | ✔ | ✗ | ✗ | ✗ | 10.0 |
| DCLM-Pool [30] | ✔ | ✔ | ✔ | ✗ | ✔ | 340.0 |
| Dolma v1.7 [52] | ✔ | ✔ | ✗ | ✔ | ✗ | 4.5 |
| Pile [17] | ✔ | ✔ | ✗ | ✔ | ✗ | 0.8 |
| SlimPajama [51] | ✔ | ✔ | ✗ | ✔ | ✗ | 0.9 |
| ROOTS [26, 27] | ✔ | ✔ | ✗ | ✔ | ✔ | 1.6 |
| RedPajama-V1 | ✔ | ✔ | ✗ | ✔ | ✗ | 3.0 |
| RedPajama-V2 | ✔ | ✔ | ✔ | ✗ | ✔ | 270.0 |

**C4** Based on the RedPajama-V2 dataset, we present a series of ablation studies on decoder-only Transformer models with 468 million parameters, showing how the quality signals can be used to create models of varying performance on common NLP benchmarks.

The remainder of this paper is organized as follows. In Section 2, we position the RedPajama dataset in the current landscape of open pretraining datasets. Section 3 describes the details of the dataset creation process behind RedPajama-V1, as well as building the RedPajama-INCITE family of models. Section 4 proceeds to RedPajama-V2, our web-only dataset. Next to describing the data processing steps, we present dataset statistics and ablation studies. Finally, we conclude in Section 5.

## 2 Related Work

Numerous efforts have focused on constructing pretraining datasets for large language models. While some of these datasets are curated from a mix of various sources, others are exclusively derived from web data. In the realm of web-only datasets, the C4 dataset [46] was one of the first large-scale web datasets, comprising a 175B token web corpus filtered down from CommonCrawl. C4 still remains a benchmark for web dataset quality. More recently, RefinedWeb [44] and FineWeb [43] have demonstrated that web-only data can yield strong models without the need for compositing multiple domains, and have, similar to our work, also provide ample details on their data curation techniques. In contrast to these datasets, RedPajama-V2 is composed of 100 trillion tokens of raw, mostly unfiltered text. With its more than 40 quality signals for potential filtering, RedPajama-V2 promotes an entirely different approach and aims to set a new standard for future high-quality web datasets, providing a robust foundation for the next generation of high quality web datasets. Of further great relevance are the Gopher rules proposed in [45], which have been central to many of the previously mentioned open pretraining datasets.

Complementing web-only datasets, composite datasets introduce additional domains and enable broader coverage. Most notably, the Pile [17] was one of the first fully open datasets. After the release of LLaMA [57], which used seven individual subsets and multiple domains, we published RedPajama-V1 as an open source replication of the LLaMA recipe, which gained widespread adoption. Building on this, the SlimPajama dataset [51] was derived from RedPajama-V1 by further cleaning and deduplication. Similarly, the Dolma [52] dataset includes other specialized domains, such as cleaned versions of code datasets including The Stack [25], StarcoderData [31] as well as the ArXiv and StackExchange splits of RedPajama-V1. The Zyda [56] dataset goes in a similar vein and further refines open datasets, including the SlimPajama dataset derived from RedPajama-V1. Finally, the ROOTS corpus [26, 27] is also among the core open datasets spanning multiple domains and languages. Table 1 shows an overview over these open datasets and makes a comparison where each dataset stands in terms of transparency, versatility and scale.

# 3 RedPajama-V1: An open Reproduction of the LLaMA Training Data

In our first iteration of the RedPajama datasets, our primary goal was to recreate the training data documented in the LLaMA technical report [57]. To this end, we closely follow the descriptions of the original recipes. In this section, we first document our process for recreating the original LLaMA training corpus (Section 3.1). We highlight gaps in the description of the original dataset collection and describe how we choose to resolve those ambiguities. Next, we report on RedPajama-INCITE, a family of models trained on this corpus in collaboration with Oak Ridge National Lab (ORNL) (Section 3.2). We find that although the resulting models are performant at the 3B scale, there remains a gap at 7B to the original LLaMA-7B model. We hypothesize that this is partly due to the need to train with the FP16 precision. In addition, this also suggests the possibility that some salient details that went into the construction of the original LLaMA training corpus may be missing.

Table 2: Token counts for the RedPajama-V1 dataset.

| Dataset Slice | Token Count |
|---|---|
| CommonCrawl | 878B |
| C4 | 175B |
| GitHub | 59B |
| Books | 26B |
| ArXiv | 28B |
| Wikipedia | 24B |
| StackExchange | 20B |
| Total | 1.2T |

## 3.1 Data Processing Steps

Here we describe our attempt to recreate the training corpus described in the LLaMa technical report [57]. The pretraining data of the LLaMA training corpus are drawn from seven datasets: English CommonCrawl, C4, GitHub, Wikipedia, Books (Project Gutenberg and Books3), ArXiv, and Stack Exchange. Each of these datasets are given a short (approximately one-paragraph) description in the LLaMA technical report, and there are some gaps in the dataset descriptions. In this section, we detail our process for recreating each of the individual datasets, highlight the gaps in the descriptions from the LLaMA technical report, and describe our choices in resolving those ambiguities. These steps together resulted in a dataset of approximately 1.2 Trillion tokens. Table 2 summarizes these datasets and the token counts. In Table 10 in the Appendix, we further list all uncertainties encountered during the construction of the dataset.

**CommonCrawl.** The LLaMA corpus includes five CommonCrawl snapshots from 2017 to 2020, processed using the CCNet pipeline [61]. CCNet deduplicates each snapshot in shards and assigns a quality classification to the data in each snapshot. It assigns a "head," "middle," and "tail" classification to each document based on the distribution of the perplexity assigned by a 5-gram Kneser-Ney model trained on Wikipedia. Here we only keep the "head" and "middle" buckets and discard the "tail." In addition, Touvron et. al. [57] use a linear classifier trained on Wikipedia reference articles to filter out low-quality documents. The LLaMA paper does not specify which snapshots were used in the dataset or give details on the classifier.

We select the five English CommonCrawl snapshots 2019-30, 2020-05, 2021-04, 2022-5, and 2023-06, representing the first snapshot in the five years preceding the start of the project. To train the Wikipedia reference classifier, we downloaded the most recent English Wikipedia snapshot available by April 1, 2023. We extract 38M URLs from the Wikipedia snapshot and crawl 300K pages. We then use the CCNet pipeline to apply a moderate cleaning step to the Wikipedia references and train a unigram classifier using fastText. Finally, we filter out all documents with scores less than 0.25, which reduces our CommonCrawl dataset to approximately the same size as the LLaMA CommonCrawl dataset.

**C4.** The LLaMA corpus includes the C4 dataset [46] to include diverse versions of CommonCrawl. We use the `c4_en` version of C4, which is provided by Allen AI on the Hugging Face Hub [2].

**Github.** The LLaMA corpus uses the public GitHub dataset available on Google BigQuery and keeps projects that are distributed under Apache, BSD, and MIT licenses. The LLaMA corpus additionally filters low-quality files using some heuristics and deduplicates at the file level. For RedPajama-V1, we remove low-quality files using a set of filters on file length, the proportion of alphanumeric characters, and file extensions. We provide the full list of heuristics in Appendix D.

---

[2] https://huggingface.co/datasets/allenai/c4

**Wikipedia.** The LLaMA corpus uses Wikipedia dumps from June to August 2022 across 20 languages, processing the data to remove hyperlinks, comments, and other formatting boilerplate. For RedPajama-V1, we use the Wikipedia dataset available on Hugging Face Hub using the dump from 2023-03-20. This also preprocesses the data to remove hyperlinks, comments, and other boilerplate.

**Gutenberg and Books3.** The LLaMA corpus uses book corpora from the Gutenberg Project and Books3 from the Pile. We only use the PG19 subset of Gutenberg and use SimHash to remove near duplicates. We originally included Books3 as well but took it down due to copyright issues.

**arXiv.** The LLaMA corpus processes arXiv LaTeX files and removes everything before the first section, comments, inline-expanded definitions and macros, and the bibliography, following [29]. We downloaded arXiv data from Amazon S3 in the "arXiv" requester pays bucket and implemented a similar postprocessing, keeping only LaTeX source files and removing preambles, comments, bibliographies, and expanding macros.

**Stack Exchange.** The LLaMa corpus includes a dump of Stack Exchange. The data is kept from the 28 largest websites, HTML tags are removed from the text, and answers are sorted by score from highest to lowest. Similarly, we download Stack Exchange from the Internet Archive, keep only the posts from the 28 largest sites, and remove HTML tags. In addition, we group the posts into question-answer pairs and order answers by their scores.

## 3.2 The RedPajama-INCITE family of LLMs

To evaluate how well RedPajama-V1 matches the original LLaMA corpus, we train a family of LLMs of various sizes in collaboration with the Incite project on the Summit supercomputer at Oak Ridge National Lab. The RedPajama-Incite family of LLMs includes a suite of pretrained and instruction-tuned models at the 3B and 7B model sizes. In this section, we first describe the compute setup of the Summit supercomputer and the implications for the pretraining runs (Section 3.2.1). We then describe how we evaluate the model and speculate on differences in quality between these models and the LLaMA family (Section 3.2.2).

### 3.2.1 Summit Training Setup

In this section, we describe the Summit supercomputer and the engineering and pretraining challenges to training the RedPajama-Incite family of LLMs. Our language models were trained using the Summit supercomputer at Oak Ridge National Lab, a cluster containing 4608 6xV100 nodes running an IBM Power9 architecture. This setup introduced a few challenges in training modern LLMs. In the following, we discuss these challenges and describe how we overcame them.

The **IBM Power9 architecture** uses a different instruction set than most modern chipsets (i.e., Intel-, Arm-, or Apple-based chips). Modern versions of PyTorch and most of the Python stack they depend on are not pre-compiled to support the Power9 architecture (the latest version officially supported was PyTorch 1.9). To support pretraining with modern libraries, members of our team needed to recompile PyTorch from scratch and build a custom training stack for Summit. Some of these efforts are documented in more detail in the GPT-NeoX technical report [6].

As of this writing, the Summit supercomputer runs on **V100 GPUs**, which are older than A100 or H100 GPUs typically used to train LLMs. Critically, V100s do not support the bf16 data type, which is necessary for modern stable training recipes for LLMs. Instead, we had to train with fp16 and use loss scaling [37] to allow for stable training runs. We also had to lower the learning rate compared to those reported in the LLaMA training, which may have had an effect on convergence ($1.6 \cdot 10^{-4}$ for the 3B model and $1.2 \cdot 10^{-4}$ for the 7B model).

The IBM Power9 architecture had **slow interconnect**, limiting the number of nodes we could use for each run. We were also unable to use the entire cluster since other projects were running simultaneously on it. We used 512 nodes in parallel (3072 GPUs) to train the 7B and 256 nodes in parallel (1536 GPUs) to train the 3B, with a global batch size of 4M tokens for each model. In scaling experiments, we found that we could not further increase the amount of parallelism without increasing the global batch size, which would hurt convergence.

The **6xV100 nodes** introduce challenges for training with tensor and pipeline parallelism. We used 12-way pipeline parallelism for the 7B and 6-way for the 3B model, as well as 2-way tensor parallelism for both models.

After accounting for these challenges, we were able to train the 3B model for 800B tokens total and the 7B model for 1.001T tokens total on Summit. We decayed the learning rate linearly following a warmup period, matching those described in the original LLaMA paper.

### 3.2.2 Evaluation

Here, we discuss evaluations for the RedPajama-INCITE-3B and 7B models on common benchmarks. The full results and benchmark scores are provided in Appendix D.2. After training RedPajama-Base-INCITE-3B for 800B tokens, it has better few-shot performance (measured in HELM classic [9]), as the average score over 16 core scenarios) and better zero-shot performance (measured using Eleuther AI's LM evaluation harness [18]) compared to other open models of similar size, including the well-regarded GPT-Neo and Pythia-2.8B (trained with 420B and 300B tokens, respectively, on the Pile). On HELM, it outperforms these models by 3-5 points. On a subset of tasks from LMevaluation harness, it outperforms these open models by 2-7 points.

The RedPajama-INCITE-7B-Base model is 1.0 points behind Falcon-7B and 4.1 points behind Llama-7B on HELM-classic. We further break down the tasks and see that they lag behind only on tasks that require using logprobs, which computes the difference between the probabilities of right and wrong answers. However, the model achieves comparable average HELM scores on tasks that directly generate answers and measure quality. Since all benchmarks in the LM harness use logprobs, we see similarly lower results for this benchmark. We hypothesize this was partly due to training with FP16, which does not allow us to use larger learning rates. Furthermore, as illustrated in the previous section, there were sources of uncertainty in the construction of the training dataset which likely resulted in a slightly different dataset than what was used to train the Llama-1 model. We believe that these two factors have led to the slightly lower performance compared to the Llama models.

RedPajama-INCITE-7B-Instruct is an instruction-tuned version of the base model optimized for few-shot performance by training on a diverse collection of NLP tasks from both P3 (BigScience) [49] and Natural Instructions (AI2) [39]. The Instruct version shows excellent performance on few-shot tasks, outperforming leading open models of similar sizes, including Llama-7B, Falcon-7B (both base and instruct version), and MPT-7B (both base and instruct version) on HELM by 2-8 points. We provide the detailed evaluation scores in the supplementary material.

## 4 RedPajama-V2

In contrast to the first iteration of the RedPajama dataset, the second iteration focuses exclusively on web data and, in addition to design principles *Transparency* and *Scale*, we also put a higher emphasis on *Versatility*. Specifically, next to the goals of providing a fully transparent and open dataset, the purpose of the corpus is to serve as a foundation for creating high quality subsets. While the goal of transparency is achieved by making the dataset and its artifacts publicly available, and scale is achieved by processing large parts of the Common Crawl corpus, to follow the design principle *Versatility*, we release RedPajama V2 as a dataset that is enriched with a set of metadata that enables fast and cheap iteration for creating high quality, diverse and large datasets. In this section, we first present the data processing steps used to create the raw text data, give an overview over the quality signals available for each document, and present statistics on the dataset composition. Finally, we present ablation studies on how the quality signals can be used to create successively better datasets.

### 4.1 Data Processing Steps

RedPajama-V2 is a dataset created by processing web documents provided by the CommonCrawl foundation[3]. As web data is inherently noisy and only available as text embedded in the HTML code, it is necessary to process it to make it suitable for training LLMs. To that end, the raw data used for RedPajama-V2 undergoes a series of basic processing steps, which we explain in more detail.

### 4.1.1 Data Acquisition

The Common Crawl Archive is a vast repository of web crawl data that is freely available to the public. The corpus contains crawling results since 2013 and is updated regularly on a (bi-) monthly

---

[3]`https://commoncrawl.org/`

basis. Next to raw web data in HTML (`warc`) format, the archive also provides metadata (`wat`) and plain text data in the `wet` format. It has been the basis for numerous datasets including C4 [46], RefinedWeb [44], Dolma [52], and FineWeb [43] among others.

To create the RedPajama-V2 dataset, we used the web-extracted text (i.e., `.wet` files) from all 84 monthly snapshots between 2014 and April 2023 and passed it through the CCNet pipeline [61]. In contrast to RPv1, here we keep all perplexity buckets, and in addition to the English language, we also keep French, German, Italian, and Spanish data. We chose this pipeline due to its light processing, which aligns with our guiding principle of preserving as much information in the raw dataset as possible and allowing downstream model developers to filter the dataset. This processing step produces over 100 billion individual text documents.

### 4.1.2 Quality Signals

A central ingredient to state-of-the-art open LLMs like Llama [57, 58], Mistral [22], Falcon [2], MPT [53], or the Qwen [3] models is the large amount of high-quality data that these models are trained on. For example, Llama 3 is trained on 15 trillion carefully curated tokens. The most prominent data sources that provide the necessary scale are the crawls made publicly available by CommonCrawl. However, this raw text, which in our case is additionally processed by the CCNet pipeline, is still not ideal for direct use as LLM training data due to artifacts arising from the conversion from HTML to plain text (e.g., parsing errors, and menus), sources of generally low quality, and biases inherent to the distribution of content on the web. To clean such datasets, the literature has proposed a multitude of heuristics to extract high-quality datasets out of large corpora of heterogeneous web data. However, unlike previous datasets that filter out low-quality content, our approach retains the entire raw text corpus, incorporating quality signals as additional metadata. This strategy allows us to use the full spectrum of data, transforming sections typically discarded into informative attributes that enhance our dataset's utility. This enables the creation of other datasets such as C4 as special cases of the RedPajama-V2 dataset. For each document, we provide the quality signals used in C4 [46], Gopher [45], RefinedWeb [44], the Pretrainer's Guide [34] and DSIR [62]. These can roughly be categorized into quality signals which measure *natural language*, the *repetiveness* of the text, are based on the *content* of the text, *ML-based* heuristics, and *deduplication*. In the following, we explain each of these categories in detail. A comprehensive list with detailed descriptions encompassing all quality signals, as well as histograms is provided in Appendix E.2.

**Natural Language.** Text documents extracted from websites often have content that does not correspond to natural language, such as JavaScript code, menus, and other boilerplate text. To measure how natural a given text document is, we provide simple heuristic measures such as the fraction of all caps words or letters, the fraction of lines that end with an ellipsis, the fraction of unique words, whether or not a line ends in a terminal punctuation mark, and others.

**Repetitiveness.** An often observed artifact of web data is repetitive text, which has been linked with uninformative content [45]. Repetitious generations are also a known failure mode of language models [21], and removing excessively repetitive content can potentially contribute to alleviating this behavior [45]. For each document, we calculate the fraction of characters appearing in the most frequent (word) $n$-gram for $n \in \{2, 3, 4\}$. Second, we calculate the fraction of characters appearing in any duplicated $n$-gram for values of $n \in \{5, \ldots, 10\}$. We ensure not to count characters that appear in overlapping $n$-grams more than once.

**Content-based.** Web documents can contain harmful and offensive content, which needs to be addressed. To that end we provide the signals used in C4 and RefinedWeb, namely, (1) the number of sequences of words that are contained in the LDNOOBW blocklist[4]. In addition, we include a flag which indicates whether the domain of the document appears in the UT1 list of blocked urls[5]. While these quality signals focus on NSFW content, we believe other content-based filters such as domains or embedding clusters [55] are also promising directions. In Figure 8 in the Appendix, we show the distribution of topics found via clustering of embeddings.

**ML Heuristics.** ML-based quality signals revolve around the idea of measuring similarity to a high-quality domain. Here, we use fastText classifiers [24], and the importance weights proposed in [62]. While ML filters have been shown to improve the quality of datasets (e.g., [12, 57, 11]), they

---

[4]`https://github.com/LDNOOBW/List-of-Dirty-Naughty-Obscene-and-Otherwise-Bad-Words`
[5]`https://dsi.ut-capitole.fr/blacklists/`

Table 3: Document and token counts for each partition and language of the RPv2 dataset.

| | All | | tail | | head+middle | | head+middle (dedupe) | |
|---|---|---|---|---|---|---|---|---|
| | docs (B) | tokens (T) | docs (B) | tokens (T) | docs (B) | tokens (T) | docs (B) | tokens (T) |
| English | 87.5 | 90.5 | 63.0 | 53.6 | 24.5 | 37.0 | 14.5 | 20.5 |
| German | 8.6 | 10.3 | 5.9 | 6.2 | 2.7 | 4.1 | 1.9 | 3.0 |
| French | 6.7 | 8.5 | 4.5 | 4.8 | 2.2 | 3.7 | 1.6 | 2.7 |
| Spanish | 6.9 | 9.5 | 4.7 | 5.6 | 2.3 | 3.9 | 1.8 | 2.8 |
| Italian | 3.5 | 4.7 | 2.4 | 2.7 | 1.2 | 1.9 | 0.9 | 1.5 |
| Total | 113.3 | 123.7 | 80.5 | 73.0 | 32.8 | 50.7 | 20.8 | 30.4 |

have also been reported to lead to biases or underrepresent minorities [15]. The fastText classifier signals provided in RPv2 are unigram bag-of-word models trained to distinguish between unfiltered RPv2 data and a high-quality domain. For English data, we use Wikipedia, websites referenced by Wikipedia, books, and the OpenWebText dataset. For non-English data, we only use Wikipedia. The DSIR weights proposed in [62] estimate the importance of individual samples to a given target domain in a reduced feature space and are based on word unigrams and bigram models. The weights are defined as the log-likelihood ratio between a language model of the target vs. the source domain, where we use the same domains as for the fasttext classifiers.

**Deduplication.** Removing duplicated training data has been found to improve model perplexity and reduce the amount of memorization while reducing the training data size and the required compute [28]. Deduplication is also one of the core building blocks of the most popular datasets [46, 52, 44]. In RPv2, we include MinHash signatures for fuzzy deduplication [10] at different similarity levels, as well as IDs of documents found to be exact duplicates via a Bloom filter [7] with the error rate set to $1\%$[6]. For this document-level deduplication, we proceed sequentially, starting with the most recent dump (2023-14) and successively iterating over the following dumps until we reach the oldest one (2014-15). An overview over how many documents were flagged as duplicates in this manner, can be seen in Figure 3 in the Appendix.

## 4.2 Dataset Statistics

RPv2 consists of 113B documents in five different languages: English, German, French, Spanish and Italian. As mentioned previously, the CCNet pipeline partitions the dataset into the tree buckets "head", "middle", and "tail" corresponding to documents with low, medium, and high Wikipedia perplexity. There are 32.8B documents in the head+middle partition and 80B documents in the tail partition. Documents in the tail are typically shorter (850 tokens) than in the head and middle buckets ($\sim$ 1500 tokens). Token counts were estimated based on an i.i.d. sample of 100M documents using the Mistral [22] BPE tokenizer. A detailed overview of token counts for each language and partition is given in Table 3. We provide further statistics on the number of documents before and after deduplication, as well as the distribution of quality signals in the supplementary materials.

## 4.3 Dataset Ablations

Here we present a series of dataset ablations with the aim of developing a better understanding of how the quality signals introduced in Section 4.1.2 influence the downstream performance of language models trained on data filtered with different heuristics. More specifically, here we ask *how do different quality filtering rules affect downstream performance?* We strive for a broad evaluation and measure the performance on diverse downstream benchmarks and the language modeling objective on multiple domains.

### 4.3.1 Setup

**Models.** We adopt decoder-only Llama-2 architectures [58] with 468M parameters and 1.6B parameters and 2048 sequence length. Both models have 24 layers, 16 attention heads and the MLP

---

[6]We remark that exact deduplication was performed based on the hashes of the `.wet` documents, i.e., prior to processing the data with CCNet.

Table 5: Evaluations for the 468M parameter LM for different dataset filters and other SOTA web datasets. The Benchmark scores are aggregated from the benchmarks outlined in Table 3, using (1) the average accuracy, (2) the Rank-Score, and (3) the normalized average score. The best score is indicated in **bold underlined** font, the second-best is **bolded**, and the third is in *italics underlined*.

| Dataset | Deduplication | | Rule-based | | ML Heuristics | | | Agg. BM-Eval (↑) | | | Val-Perplexity (↓) | |
|---|---|---|---|---|---|---|---|---|---|---|---|---|
| | Exact | Fuzzy | C4 | Gopher | Classif. | DSIR | PPL | Avg. | Norm. Avg. | Rank-Score | Pile | Paloma |
| C4 | | | | | | | | 35.8 | 0.140 | 0.472 | 29.5 | 39.5 |
| Dolma-v1.7 CC | | | | | | | | 36.0 | 0.140 | 0.511 | 21.4 | 38.3 |
| FineWeb | | | | | | | | 36.5 | 0.146 | _0.644_ | 26.8 | 33.6 |
| RefinedWeb | | | | | | | | **37.9** | **0.165** | **0.650** | _19.1_ | _32.8_ |
| RPv1-CC | ✔(sharded) | | | | ✔ (Wiki-Ref.) | | | 35.6 | 0.127 | 0.461 | 18.7 | 31.5 |
| RPv2 (2023-14) | ✔ | | | | | | | 36.4 | 0.141 | 0.594 | 19.7 | **31.1** |
| RPv2 (2023-14) | | ✔ | | | | | | 36.2 | 0.138 | 0.472 | 19.5 | 39.9 |
| RPv2 (2023-14) | | ✔ | ✔ | ✔ (full) | | | | **37.6** | **0.160** | **0.700** | 24.9 | 34.5 |
| RPv2 (2023-14) | | ✔ | ✔ | | | | | 36.8 | 0.150 | 0.622 | 36.3 | 56.9 |
| RPv2 (2023-14) | | ✔ | | ✔ (natlang) | | | Wiki-middle | 37.2 | 0.154 | 0.639 | 23.6 | 38.2 |
| RPv2 (2023-14) | | ✔ | | ✔ (Rep.) | | | Wiki-middle | _37.5_ | _0.158_ | 0.633 | 20.4 | 36.0 |
| RPv2 (9 Dumps) | | ✔ | ✔ | | | | | 35.3 | 0.128 | 0.517 | 35.0 | 54.2 |
| RPv2 (9 Dumps) | | ✔ | ✔ | ✔ (full) | | | | 36.7 | 0.149 | 0.556 | 43.8 | 63.9 |
| RPv2 (9 Dumps) | | ✔ | ✔ | ✔ (Rep.) | | ✔ (Palm-mix) | | 35.9 | 0.138 | 0.439 | 44.3 | 89.9 |
| RPv2 (9 Dumps) | | ✔ | ✔ | ✔ (Rep.) | ✔ (Palm-mix) | | | 35.9 | 0.139 | 0.483 | 43.8 | 67.1 |
| RPv2 (9 Dumps) | | ✔ | ✔ | ✔ (natlang) | ✔ (Palm-mix) | | | 36.7 | 0.152 | 0.550 | 41.8 | 67.9 |
| RPv2 (9 Dumps) | | ✔ | ✔ (line-filter) | ✔ (natlang) | ✔ (Palm-mix) | | | 36.4 | 0.144 | 0.539 | 32.4 | 52.9 |
| RPv2 (9 Dumps) | | ✔ | custom-rules | | ✔ (Wiki-Ref.) | | $P_{wiki} > 30$ | 35.8 | 0.130 | 0.467 | **18.5** | 39.7 |
| RPv2 (9 Dumps) | | ✔ | custom-rules + Gopher-Rep. | | ✔ (Wiki-Ref.) | | $P_{wiki} > 30$ | 35.9 | 0.133 | 0.500 | 19.8 | 45.8 |

expansion ratio set to 4.0. The 468M has 1024 hidden dimension, while for the 1.6B we use 2048. For each dataset, we train the 468M model on 100B tokens and the 1.6B model on 350B tokens. We use the AdamW [14] optimizer with the weight decay of 0.1, a maximum learning rate set to $5e-3$ and $5e-4$ respectively, and a cosine decay schedule with linear warmup during the first 1% of steps. We use relatively small scales, as this enables us to explore a wider range of filters, showing the breadth of the quality filters available in RedPajama.

**Hardware and Training Stack.** Due to its ease of setup, use, and high model flops utilization, we use the OLMo framework[7] for distributed training, using FSDP [66] for parallelization across multiple GPUs and nodes. For evaluation, we use the lm-evaluation-harness. We train our models on up to 5 H100 nodes with Infiniband interconnect.

**Evaluation Metrics.** We strive for broad coverage of benchmarks and domains. At the same time, we are operating at a relatively small scale, where many tasks are too hard to provide a high enough signal to distinguish datasets.

Similar to the FineWeb [43] dataset, we look for benchmarks that provide a high enough signal-to-noise ratio, even at this small model scale. After careful consideration, we settled for the choice of benchmarks in Table 4. Here we present aggregated scores by (1) computing the average over benchmarks, (2) the normalized average, and (3) the normalized sum of ranks for each data recipe. We chose to include the latter to avoid averaging over scores with different scales. More detailed scores are provided in the supplementary materials. To rank datasets based on the Perplexity of target domains, we follow the approach taken in Dolma [52] and adopt the Paloma [35] and Pile [17] validation sets.

Table 4: Benchmarks used in our ablations. The column "Agg. BM-Eval" indicates whether the score is used in the aggregate scores reported in Tables 5 and 6.

| Task | Type | Random | Metric | Agg. BM-Eval |
|---|---|---|---|---|
| ANLI [40] | Natural language inference | 25.0 | acc | |
| ARC-c [13] | Natural language inference | 25.0 | acc_norm | ✔ |
| ARC-e [13] | Natural language inference | 25.0 | acc_norm | ✔ |
| Winogrande [48] | Coreference resolution | 50.0 | acc | ✔ |
| Hellaswag [64] | Sentence completion | 25.0 | acc_norm | ✔ |
| LAMBADA [42] | Sentence completion | 0.0 | acc | ✔ |
| CoQA [47] | Conversational QA | 0.0 | F1 | ✔ |
| MMLU [20] | Multiple-choice QA | 25.0 | acc | ✔ |
| OpenbookQA [38] | Multiple-choice QA | 25.0 | acc_norm | ✔ |
| PIQA [5] | Multiple-choice QA | 50.0 | acc_norm | ✔ |
| PubMedQA [23] | Multiple-choice QA | 33.3 | acc | ✔ |
| SciQ [60] | Multiple-choice QA | 25.0 | acc_norm | ✔ |
| SocialIQA [50] | Multiple-choice QA | 25.0 | acc | |
| TruthfulQA [33] | Multiple-choice QA | 25.0 | acc | |

### 4.3.2 Results

We start by using the quality signals to implement some of the most widely used filters in the literature. In addition, we also investigate ML heuristics available in RPv2, which are based on fastText bag of $n$-gram classifiers [24] and DSIR importance weights [62]. We run ablations on two subsets of RPv2, namely on the 2023-14 crawl and on the 9 crawls from 2021-49 to 2023-14, which were deduplicated using MinHash LSH on word 13-grams, with 128 hash functions, 9 bands and 13 rows. For the 1.6B

---

[7] https://github.com/allenai/OLMo

Table 6: Aggregated evaluations for the 1.6B parameter LM for different datasets. The Benchmark scores are aggregated from the benchmarks outlined in Table 4, using (1) the average accuracy, (2) the Rank-Score, and (3) the normalized average score.

| Dataset | Fuzzy Deduplication | Rule-based | | ML Heuristics | | Agg. BM-Eval (↑) | | | Val-Perplexity (↓) | |
|---|---|---|---|---|---|---|---|---|---|---|
| | | C4 | Gopher | Palm Classif. | Wiki-Ref Classif. | Avg. | Norm. Avg. | Rank-Score | Pile | Paloma |
| RefinedWeb | | | | | | 52.0 | 34.0 | 0.139 | 10.7 | 17.7 |
| RPv2 (full) | ✔ | | ✔ | | ✔ | 50.0 | 31.1 | 0.106 | 13.6 | 20.8 |
| RPv2 (full) | ✔ | ✔ | ✔(natlang) | ✔ | | 47.9 | 29.4 | 0.089 | 22.2 | 30.7 |

ablations, we filter the full RPv2 dataset, then sample roughly 1T tokens and deduplicate it with the same Minhash hyperparameters.

**Filters.** We seek to cover a wide range of quality filtering configurations. Rather than optimizing the performance on a particular benchmark, the goal is to show that filtering the RPv2 dataset in different ways can lead to wildly different model performance. We thus experiment with variations of the C4 and Gopher rules and also use the ML-based quality signals in RPv2. We also use a custom configuration `custom-rules` based on word counts, average line length, Wikipedia Perplexity and the Wikipedia References classifier.

**Results.** From Table 5, we can draw a series of conclusions on filtering the RedPajama-V2 dataset. *First*, we can see that the Gopher rules generally improve performance. In particular we see that fuzzy deduplication and filtering with Gopher has the highest aggregated scores across all RPv2 datasets. In addition, both the average and normalized average benchmark score is only second to RefinedWeb, while the rank-score is higher than for RefinedWeb. The per-benchmark tables 18, 19, and 20 in the Appendix, show that the RPv2 dataset filtered with fuzzy deduplication and Gopher is always in the upper middle (minimum rank score 9 of 19), while RefinedWeb is performing worse on Hellaswag, LAMBADA, Winogrande, MMLU and OpenBookQA. This indicates that filtering RPv2 with the full Gopher rules and fuzzy deduplication (Minhash LSH) creates a dataset that performs well across a wider range of tasks than all other datasets. *Second*, we can see that the Gopher-natlang filters perform better than the Gopher-repetition filters. *Third*, in the context of model based filtering, we see no significant difference between using a fasttext classifier and DSIR. *Fourth*, using only the line-level C4 filters appears to reduce perplexity, but has negligible effect on the aggregated benchmark scores. Finally, we notice that the unfiltered RPv2 2023-14 dataset appears to have the lowest perplexity on the Paloma dataset, while other filtering methods lead to models with higher perplexity. We believe that this can (at least in part) be attributed to the wide range of domains covered by Paloma. In addition, Paloma also contains the RPv1 dataset, which can explain the low Perplexity score obtained by the model trained on RPv1-CC. Table 6 shows further that the model trained on RPv2 filtered with the full Gopher rules outperforms the model trained on RPv2 filtered with only the Gopher-natlang rules, and comes close to the quality of a model trained on the RefinedWeb dataset. In conclusion, this series of ablation studies shows how the quality signals in the RPv2 dataset can be used to successively filter better datasets. In combination with its vast scale of over 100T tokens, we see that this dataset provides a powerful source for creating high-quality web datasets for LLM pretraining.

## 5 Conclusion

In this paper, we have presented the RedPajama datasets. With over 100 Trillion tokens, these are the largest, fully open, and transparent datasets for pretraining language models and have been a central building block for many of the strongest open-source LLMs. Next to documentation accompanying the datasets, we have also shown examples of how RedPajama-V2 can be filtered down to successively higher quality subsets, leading to language models of varying levels of quality on a diverse set of benchmark tasks and outperforming models trained on other large-scale pretraining corpora. While the models are relatively small and enabled us to explore a wider variety of filters, it is also a limitation and further, larger-scale explorations are required. We did not explore a thorough decontamination analysis against common benchmarks or an analysis of personally identifiable information present in the dataset, posing another limitation of this work. By publishing the RedPajama-V2 dataset in raw, unfiltered form, but accompanied by a set of quality signals, we hope that future work will continue to build on RedPajama and provide new innovative ways of filtering, curating, and mixing multiple pretraining corpora.

## Acknowledgments and Disclosure of Funding

We acknowledge support from the Canada CIFAR AI Chair Program [I.R.], and the Canada Excellence Research Chairs Program [I.R.]. This research was made possible thanks to the computing resources on the Summit supercomputer, provided as a part of the INCITE 2023 program award "Scalable Foundation Models for Transferable Generalist AI". These resources were provided by the Oak Ridge Leadership Computing Facility at the Oak Ridge National Laboratory, which is supported by the Office of Science of the U.S. Department of Energy under Contract No. DE-AC05-00OR22725. We gratefully acknowledge the support of NIH under No. U54EB020405 (Mobilize), NSF under Nos. CCF2247015 (Hardware-Aware), CCF1763315 (Beyond Sparsity), CCF1563078 (Volume to Velocity), and 1937301 (RTML); US DEVCOM ARL under Nos. W911NF-23-2-0184 (Long-context) and W911NF-21-2-0251 (Interactive Human-AI Teaming); ONR under Nos. N000142312633 (Deep Signal Processing); Stanford HAI under No. 247183; NXP, Xilinx, LETI-CEA, Intel, IBM, Microsoft, NEC, Toshiba, TSMC, ARM, Hitachi, BASF, Accenture, Ericsson, Qualcomm, Analog Devices, Google Cloud, Salesforce, Total, the HAI-GCP Cloud Credits for Research program, the Stanford Data Science Initiative (SDSI), and members of the Stanford DAWN project: Meta, Google, and VMWare. The U.S. Government is authorized to reproduce and distribute reprints for Governmental purposes notwithstanding any copyright notation thereon. Any opinions, findings, and conclusions or recommendations expressed in this material are those of the authors and do not necessarily reflect the views, policies, or endorsements, either expressed or implied, of NIH, ONR, or the U.S. Government.

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

# A   Appendix

## A.1   Checklist


## B   Intended Uses

The RedPajama datasets were created with the primary use case of providing training data for large language models. RedPajama contains data from different sources and domains. RedPajama-V1 contains data obtained from web scrapes, Wikipedia articles, scientific content extracted from articles available on arXiv, as well as code from various programming languages. RedPajama-V2 contains data exclusively based on web scrapes and is accompanied by a series of quality signals that are intended to be used for filtering the raw dataset.

## C   Dataset Accessibility

Both RedPajama-V1 and RedPajama-V2 are available for download via the Huggingface Hub at `https://huggingface.co/datasets/togethercomputer/RedPajama-Data-1T` and `https://huggingface.co/datasets/togethercomputer/RedPajama-Data-V2`.

**Access via public HTTP endpoints.** We also provide access to the datasets via public HTTPS endpoints. The list of urls for components of the RedPajama-V1 dataset can be obtained from `https://data.together.xyz/redpajama-data-1T/v1.0.0/urls.txt`. The list of urls for the different components of RedPajama-V2 can be obtained from the following urls:

- Raw text documents can be obtained from `https://data.together.xyz/redpajama-data-v2/v1.0.0/urls/document-urls.txt`
- Quality signals for the `head` and `middle` partitions can be obtained from `https://data.together.xyz/redpajama-data-v2/v1.0.0/urls/quality_signals-urls.txt`
- The list of document ids which are exact duplicates can be obtained from `https://data.together.xyz/redpajama-data-v2/v1.0.0/urls/duplicates-urls.txt`
- The minhash signatures can be obtained from `https://data.together.xyz/redpajama-data-v2/v1.0.0/urls/minhash-urls.txt`

### C.1   Structure of the datasets

Both RedPajama-V1 and RedPajama-V2 are distributed in the JSON Lines format and partitioned into shards. Due to their differing nature, the two datasets are structured differently.

#### C.1.1   RedPajama-V1

RedPajama-V1 consists of seven domains and is structured accordingly. Except for the Common Crawl subset, each component follows the structure

```
{
    "text": "...", "meta": {...}
}
```

The `meta` field varies between the different sources:

- The **Arxiv** subset has the meta fields `timestamp`, `yymm`, `arxiv_id`, `language` and `url`.
- The **C4** subset has the meta fields `timestamp`, `source`, `language` and `url`.
- The **Github** subset has the meta fields `content_hash`, `timestamp`, `source`, `line_count`, `max_line_length`, `avg_line_length`, `alnum_prop`, `repo_name`, `id`, `size`, `binary`, `copies`, `ref`, `path`, `mode`, `license`, `language`.
- The **Stack Exchange** subset has the meta fields `timestamp`, `source`, `language`, `question_score` and `url`.
- The **Wikipedia** subset has the meta fields `timestamp`, `title`, `language`, and `url`.

The Common Crawl subset follows the structure

```
{
    "text": "...",
```

```
    "pred_label": ...,
    "pred_label_prob": ...,
    "wiki_prob": ...,
    "source": "..."
}
```

### C.1.2 RedPajama-V2

The core of the dataset is composed of the text documents, accompanied by the quality annotations, duplicate ids and minhash signatures. For the text documents, the structure largely follows the one defined by CCNet. Specifically, the documents for a given CommonCrawl snapshot are partitioned into 5000 shards, where the filename indicates the shard, language of the document, and the perplexity bucket (partition). The quality annotations, duplicates and minhashes follow the same logic and reflect the filenames of the raw documents.

**File Structure.** The files containing the raw text documents are organized according to the following pattern:

documents/<snapshot_id>/<shard_id>/<lang>_<ppl_bucket>.json.gz

where `snapshot_id` corresponds to any of the crawls included in RPv2, `shard_id` ranges from 0000 to 4999, `lang` is any one of en, de, fr, es or it. Finally, `ppl_bucket` indicates the partitioning according to Wikipedia perplexity and is either head, middle or tail. Similarly, quality signals, duplicate ids and minhashes follow the patterns

quality_signals/<snapshot_id>/<shard_id>/<lang>_<ppl_bucket>.signals.json.gz,
 duplicates/<snapshot_id>/<shard_id>/<lang>_<ppl_bucket>.duplicates.parquet,

and

minhashes/<snapshot_id>/<shard_id>/<lang>_<ppl_bucket>.minhash.parquet.

**Documents Structure.** The `documents` are stored as Gzip-compressed JSONL files and follow the schema

```
{
    "url": "...",
    "date_download": "...",
    "digest": "...",
    "length": ...,
    "nlines": ...,
    "source_domain": "...",
    "title": "...",
    "raw_content": "...",
    "cc_segment": "...",
    "original_nlines": ...,
    "original_length": ..,
    "line_ids": [...],
    "language": "...",
    "language_score": ...,
    "perplexity": ...,
    "bucket": "..."
}
```

**Quality Signals Structure.** The `quality signals` are Gzip-compressed JSONL files and follow the schema

```
{
    "id": "...",
    "id_int": ...,
    "metadata": {
        "cc_segment": "...",
```

```
        "cc_net_source": "...",
        "url": "...",
        "source_domain": "...",
        "language": "...",
        "snapshot_id": "..."
    },
    "quality_signals": {
        "key": [[start, end, score]]
    }
}
```

The `quality_signals` field is a dictionary with the name of the quality signal as key and a list of tuples as values. Each tuple consists of the three floats `start`, `end`, and `score` indicating the location where the `score` in the `raw_content` string applies. This representation follows the one used in Dolma [52] and allows a single representation to encode quality signals which apply at different levels of granularity of the text (e.g., line-level and document-level).

**Duplicate IDs.** The ids of duplicated documents are stored as parquet files. Each row in the parquet file corresponds to a document which is duplicated at least once across the entire corpus. We emphasize that this does not include the first occurrence of a document which has subsequent duplicates. In other words, if every document appearing in the list of duplicates is dropped, one member of each cluster of documents remains in the dataset.

**Minhashes.** The minhash signatures are stored in parquet files and are partitioned into multiple bands and rows, corresponding to different levels of Jaccard similarities in the range $\{0.7, 0.8, 0.9, 1.0\}$.

## D  RedPajama-V1

Here we provide additional details and results for the RedPajama-V1 dataset.

### D.1  Filtering Heuristics for Code obtained from GitHub

As indicated in the main part of this paper, we filter the raw GitHub dataset by keeping only projects under Apache, BSD and MIT licenses, and additionally apply filtering heuristics similar to the ones used in The Stack dataset [25]. Specifically, we apply the following set of heuristics, removing any file with the following properties:

- maximum line length of more than 1000 characters.
- an average line length of more than 100 characters.
- a proportion of alphanumeric characters of less than 0.25.
- a ratio between the number of alphabetical characters and the number of tokens of less than 1.5.
- extension is not in the following set of whitelisted extensions: .asm, .bat, .cmd, .c, .h, .cs, .cpp, .hpp, .c++, .h++, .cc, .hh, .C, .H, .cmake, .css, .dockerfile, .f90, .f, .f03, .f08, .f77, .f95, .for, .fpp, .go, .hs, .html, .java, .js, .jl, .lua, .md, .markdown, .php, .php3, .php4, .php5, .phps, .phpt, .pl, .pm, .pod, .perl, .ps1, .psd1, .psm1, .py, .rb, .rs, .sql, .scala, .sh, .bash, .command, .zsh, .ts, .tsx, .tex, .vb, Dockerfile, Makefile, .xml, .rst, .m, .smali

### D.2  Detailed Evaluations for the RedPajama-INCITE LLMs

Here we provide detailed benchmark scores for the RedPajama-INCITE 3B and 7B LLMs, trained on the RedPajama-V1 dataset.

### D.3  Detailed Sources of Uncertainties in the Construction of the RedPajama-V1 Dataset

Table 10 shows a detailed overview over different sources of uncertainty that arose during the construction of the RedPajama-V1 dataset. These uncertainties mainly stem from a lack of details on the dataset presented in [57]. From this list, it can be seen that it is likely that there is a mismatch

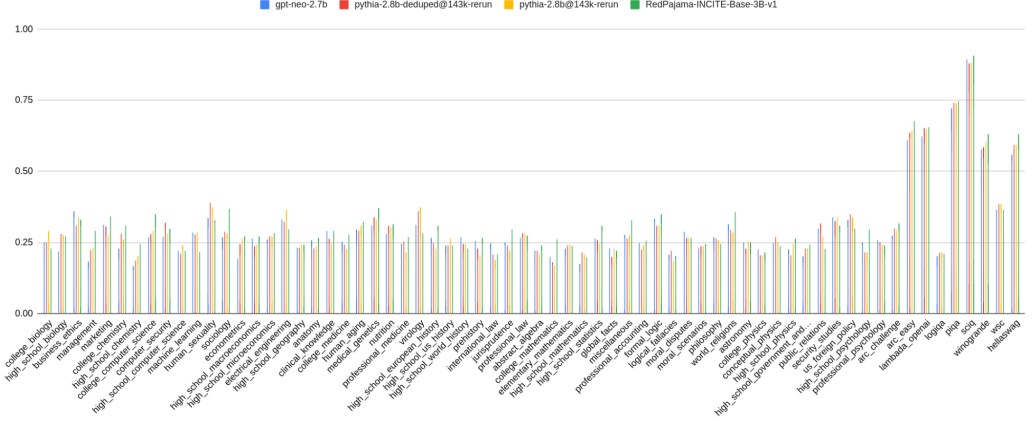

Figure 2: RedPajama-INCITE-Base 3B results on a subset of lm-evaluation-harness. The tasks were selected according to the selection made to evaluate Pythia [4] and GPT-J [59]

Table 7: Results for RedPajama-INCITE-Base-3B-v1 on a subset of lm-evaluation-harness (Zero-Shot) and HELM, compared to models with similar parameter counts. The top-scoring model for each benchmark is highlighted in **bold** font.

|  | Lambada OpenAi (acc) | Hellaswag (acc_norm) | Winogrande (acc) | Piqa (acc) | Avg. | HELM avg. |
|---|---|---|---|---|---|---|
| GPT-Neo | 0.6223 | 0.5579 | 0.5769 | 0.7219 | 0.6197 | 0.3570 |
| Pythia-2.8B | 0.6466 | 0.5933 | 0.6006 | 0.7399 | 0.6451 | 0.3770 |
| Pythia-2.8B-dedup | **0.6524** | 0.5941 | 0.5848 | 0.7404 | 0.6429 | - |
| RedPajama-INCITE-Base-3B-v1 | 0.6541 | **0.6317** | **0.6322** | **0.7470** | **0.6662** | **0.4060** |

Table 8: HELM Benchmark results for RedPajama-INCITE-Base-7B-v1 and instruction tuned. The top-scoring model for each benchmark is highlighted in **bold** font.

| Model | RedPajama 7B Instruct | Llama 7B | MPT 7B | Falcon 7B | RedPajama 7B Base | GPT J | Falcon 7B Instruct | Pythia 7B | Dolly v2 | MPT 7B Instruct | Stablelm Alpha 7B |
|---|---|---|---|---|---|---|---|---|---|---|---|
| HELM-AVG | **0.492** | 0.472 | 0.444 | 0.441 | 0.431 | 0.417 | 0.407 | 0.400 | 0.396 | 0.393 | 0.288 |
| MMLU - EM | **0.366** | 0.345 | 0.294 | 0.285 | 0.323 | 0.249 | 0.271 | 0.266 | 0.238 | 0.349 | 0.293 |
| BoolQ - EM | 0.697 | 0.751 | 0.731 | **0.770** | 0.694 | 0.649 | 0.708 | 0.656 | 0.602 | 0.442 | 0.537 |
| NarrativeQA - F1 | **0.623** | 0.524 | 0.541 | 0.549 | 0.512 | 0.545 | 0.381 | 0.427 | 0.441 | 0.220 | 0.218 |
| NaturalQuestions (closed-book) - F1 | 0.229 | **0.297** | 0.284 | 0.289 | 0.258 | 0.156 | 0.192 | 0.141 | 0.133 | 0.247 | 0.077 |
| NaturalQuestions (open-book) - F1 | **0.654** | 0.580 | 0.603 | 0.574 | 0.600 | 0.559 | 0.453 | 0.549 | 0.535 | 0.627 | 0.317 |
| QuAC - F1 | 0.252 | 0.332 | 0.343 | 0.322 | 0.323 | 0.330 | 0.300 | 0.306 | 0.299 | **0.352** | 0.218 |
| *HellaSwag - EM | 0.698 | 0.747 | 0.754 | 0.732 | 0.702 | 0.663 | 0.690 | 0.653 | 0.692 | **0.763** | 0.421 |
| *OpenbookQA - EM | 0.488 | 0.574 | 0.540 | **0.546** | 0.504 | 0.514 | 0.498 | 0.496 | 0.516 | 0.532 | 0.394 |
| TruthfulQA - EM | 0.226 | 0.297 | 0.186 | 0.206 | 0.205 | 0.199 | 0.203 | 0.225 | **0.250** | 0.188 | 0.209 |
| *MS MARCO (regular) - RR@10 | **0.391** | 0.252 | 0.161 | 0.169 | 0.135 | 0.152 | 0.225 | 0.159 | 0.160 | 0.161 | 0.110 |
| *MS MARCO (TREC) - NDCG@10 | **0.709** | 0.482 | 0.369 | 0.362 | 0.322 | 0.345 | 0.481 | 0.342 | 0.359 | 0.387 | 0.253 |
| CNN/DailyMail - ROUGE-2 | 0.143 | **0.149** | 0.137 | 0.147 | 0.137 | 0.131 | 0.114 | 0.101 | 0.140 | 0.148 | 0.045 |
| XSUM - ROUGE-2 | 0.101 | **0.127** | 0.107 | 0.116 | 0.114 | 0.096 | 0.071 | 0.079 | 0.074 | 0.101 | 0.037 |
| IMDB - EM | 0.941 | 0.933 | 0.903 | 0.893 | 0.916 | **0.939** | 0.906 | 0.930 | 0.907 | 0.891 | 0.627 |
| CivilComments - EM | **0.667** | 0.578 | 0.525 | 0.511 | 0.536 | 0.520 | 0.516 | 0.527 | 0.520 | 0.270 | 0.490 |
| RAFT - EM | 0.682 | 0.583 | 0.618 | 0.586 | 0.611 | **0.619** | 0.498 | 0.542 | 0.466 | 0.616 | 0.368 |

Table 9: LM eval harness results for RedPajama-INCITE-Base-7B-v1 and instruction tuned model. The top-scoring model for each benchmark is highlighted in **bold** font.

|  | MPT 7B Instruct | Falcon 7B | MPT 7B | RedPajama 7B Base | Llama 7B | RedPajama 7B Instruct | Falcon 7B Instruct | Dolly v2 | GPT-J | Pythia 7B | StableLM Alpha 7B |
|---|---|---|---|---|---|---|---|---|---|---|---|
| LM-eval-harness-AVG | **0.7195** | 0.7161 | 0.7100 | 0.6882 | 0.6881 | 0.6858 | 0.6813 | 0.6557 | 0.6526 | 0.6392 | 0.5260 |
| arc_challenge (acc_norm) | **0.4462** | 0.4326 | 0.4215 | 0.3925 | 0.4147 | 0.4078 | 0.4283 | 0.4027 | 0.3660 | 0.3532 | 0.2705 |
| arc_easy (acc) | **0.7218** | 0.7096 | 0.7008 | 0.6923 | 0.5253 | 0.7159 | 0.6789 | 0.6423 | 0.6225 | 0.6338 | 0.4487 |
| boolq (acc) | 0.7425 | 0.7361 | **0.7486** | 0.707 | 0.7315 | 0.6865 | 0.7089 | 0.6502 | 0.6544 | 0.6446 | 0.6006 |
| copa (acc) | **0.9000** | 0.8600 | 0.8500 | 0.880 | 0.8500 | 0.850 | 0.8400 | 0.8600 | 0.8300 | 0.7400 | 0.7500 |
| hellaswag (acc_norm) | **0.7717** | 0.7634 | 0.7626 | 0.7037 | 0.7620 | 0.7103 | 0.6978 | 0.6896 | 0.6625 | 0.6588 | 0.4122 |
| lambada_openai (acc) | 0.6918 | **0.7467** | 0.7056 | 0.7143 | 0.7360 | 0.6895 | 0.6831 | 0.6893 | 0.6831 | 0.6441 | 0.6379 |
| piqa (acc_norm) | 0.8041 | **0.8069** | 0.8052 | 0.7737 | 0.7810 | 0.7699 | 0.7856 | 0.7486 | 0.7617 | 0.7671 | 0.6736 |
| winogrande (acc) | 0.6780 | 0.6732 | **0.6859** | 0.6417 | 0.7040 | 0.6567 | 0.6669 | 0.6140 | 0.6409 | 0.6267 | 0.5012 |

between RedPajama-V1 and the dataset used to train the Llama-1 models. We believe this is a significant factor that has contributed to the performance mismatch between RedPajama-INCITE and LLaMA-1.

Table 10: Overview over the different uncertainties and decisions made during the construction of the RedPajama-V1 dataset.

| Subset | Uncertainty | Decision |
|---|---|---|
| CommonCrawl | Which snapshots were used? | We use the first snapshot from 2019 to 2023. |
| | What classifier was used, and how was it constructed? | We use a fasttext classifier with unigram features and use 300k training samples. |
| | What threshold was used to classify a sample as high quality? | We set the threshold to match the token count reported in LLama. |
| GitHub | Quality filtering heuristics | We remove any file
• with a maximum line length of more than 1000 characters.
• with an average line length of more than 100 characters.
• with a proportion of alphanumeric characters of less than 0.25.
• with a ratio between the number of alphabetical characters and the number of tokens of less than 1.5.
• whose extension is not in the following set of whitelisted extensions: .asm, .bat, .cmd, .c, .h, .cs, .cpp, .hpp, .c++, .h++, .cc, .hh, .C, .H, .cmake, .css, .dockerfile, .f90, .f, .f03, .f08, .f77, .f95, .for, .fpp, .go, .hs, .html, .java, .js, .jl, .lua, .md, .markdown, .php, .php3, .php4, .php5, .phps, .phpt, .pl, .pm, .pod, .perl, .ps1, .psd1, .psm1, .py, .rb, .rs, .sql, .scala, .sh, .bash, .command, .zsh, .ts, .tsx, .tex, .vb, Dockerfile, Makefile, .xml, .rst, .m, .smali |
| Wikipedia | Which Wikipedia dump was used? | We used the most recent at the time of data curation (2023-03-20). |
| Books | How were the books deduplicated? | We use SimHash to perform near deduplication. |

# E    RedPajama-V2

In this section we provide additional analyses and statistics of the RedPajama-V2 web dataset, and present detailed results for the ablation models trained on differently filtered subsets.

## E.1    Summary Statistics of our Deduplication Approach

In Figure 3, we see how the number of documents in the head+middle partition develops as a function of the point in time of each crawl. What stands out here is that there is a relatively stable number until 2018 and a significantly smaller number of documents between 2014 and 2016 (up to 10x for, e.g., German). It is also worth noting how the number of unique documents over time develops (dashed line). Specifically, since we ran the deduplication from the newest snapshot to the oldest, one expects an increasingly smaller number of unique documents in the corpus, which can be observed from Figure 3 (note the log-scale). However, it is worth pointing out the sudden drop in unique documents occurring for the crawls between 2014 and 2017. We believe that this can be explained by a different list of seeds used by the CommonCrawl web crawler during that period.

## E.2    Quality Signals

In this section we provide further details and statistics on the quality signals which are part of the RedPajama-V2 dataset.

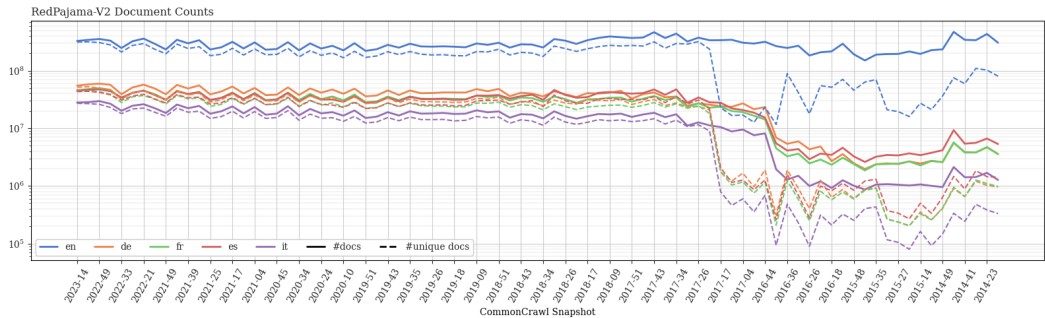

Figure 3: Chronological count of documents for each CommonCrawl snapshot before and after deduplication. Deduplication is performed sequentially, starting from the most recent snapshot and iterating until the oldest snapshot.

Table 11: Quality signals originating from the CCNet pipeline [61].

| Annotation Tag | Description |
| --- | --- |
| ccnet_bucket | head, middle or tail bucket of the perplexity score |
| ccnet_language_score | score of the language identification model |
| ccnet_length | number of characters |
| ccnet_nlines | number of lines |
| ccnet_original_length | number of characters before line-level deduplication |
| ccnet_original_nlines | number of lines before line-level deduplication |
| ccnet_perplexity | perplexity of an LM trained on Wikipedia |

### E.2.1 Overview of Available Quality Signals

The set of quality signals can be grouped into such with measure *natural language* (Table 12), the *repetitiveness* of the text (Table 14), are based on the *content* of the text (Table 15), or which are *ML-based* heuristics (Table 13). In addition, here we also summarize the quality signals which are computed by the CCNet pipeline in Table 11.

### E.2.2 Histograms

Histograms of the distribution of quality signals are shown in Figures 4,5,6 and 7. The statistics are obtained from the 2023-06 snapshot and are computed only for English data.

### E.3 Embedding-based Clustering

To compute clusters based on the semantics of text documents, we randomly sampled 2,000,000 documents from the unfiltered 2021-04 snapshot of the RedPajama-V2 dataset and used the Alibaba-NLP gte-large-en-v1.5 model [32] to compute embeddings on the middle 8,192 tokens of each document. We used Nomic Atlas [41] for clustering and topic modeling analysis. An overview of clusters and associated topics is shown in Figure 8. 6 randomly sampled documents, along with their corresponding cluster topics and a 1000-character substring from each document (starting after a random whitespace character), are shown in Table 16 and Table 17.

### E.4 Data Ablations: Detailed Evaluations

We have shown aggregated benchmark scores in the main part of this work. Here, we provide more details and report the scores for each task separately. The results are shown in Tables 18, 19 and 20.

Table 12: Summary of quality signals which measure how much a document corresponds to natural language.

| Annotation Tag | Description | Reference(s) |
|---|---|---|
| rps_doc_curly_bracket | The ratio between the number of occurrences of '{' or '}' and the number of characters in the raw text. | [46] |
| rps_doc_frac_all_caps_words | The fraction of words in the content that only consist of uppercase letters. This is based on the raw content. | [34] |
| rps_doc_frac_lines_end_with_ellipsis | The fraction of lines that end with an ellipsis, where an ellipsis is defined as either "..." or "U+2026". | [44, 45] |
| rps_doc_frac_no_alph_words | The fraction of words that contain no alphabetical character. | [44, 45] |
| rps_doc_lorem_ipsum | The ratio between the number of occurrences of 'lorem ipsum' and the number of characters in the content after normalisation. | [46] |
| rps_doc_mean_word_length | The mean length of words in the content after normalisation. | [44, 45] |
| rps_doc_stop_word_fraction | The ratio between the number of stop words and the number of words in the document. Stop words are obtained from `https://github.com/6/stopwords-json`. | [44, 45] |
| rps_doc_symbol_to_word_ratio | The ratio of symbols to words in the content. Symbols are defined as U+0023 (#), "...", and U+2026. | [44, 45] |
| rps_doc_frac_unique_words | The fraction of unique words in the content. This is also known as the degeneracy of a text sample. Calculated based on the normalised content. | [34] |
| rps_doc_unigram_entropy | The entropy of the unigram distribution of the content. This measures the diversity of the content and is computed using $\sum_x -\frac{x}{n} \cdot \log(\frac{1}{n})$ where the sum is taken over counts of unique words in the normalised content. | - |
| rps_doc_word_count | The number of words in the content after normalisation. | [44, 45] |
| rps_lines_ending_with_terminal_punctution_mark | Indicates whether a line ends with a terminal punctuation mark. A terminal punctuation mark is defined as one of: ".", "!", "?", "”". | [46] |
| rps_lines_javascript_counts | The number of occurrences of the word "javascript" in each line. | [46] |
| rps_lines_num_words | The number of words in each line. This is computed based on the normalised text. | [46, 44] |
| rps_lines_numerical_chars_fraction | The ratio between the number of numerical characters and total number of characters in each line. This is based on the normalised content. | [44] |
| rps_lines_start_with_bulletpoint | Whether the lines that start with a bullet point symbol. The following set of unicodes are considered a bullet point: U+2022 (bullet point), U+2023 (triangular bullet point), U+25B6 (black right pointing triangle), U+25C0 (black left pointing triangle), U+25E6 (white bullet point), U+2013 (en dash) U+25A0 (black square), U+25A1 (white square), U+25AA (black small square), U+25AB (white small square). | [43, 45] |
| rps_lines_uppercase_letter_fraction | The ratio between the number of uppercase letters and total number of characters in each line. This is based on the raw text. | [44] |
| rps_doc_num_sentences | The number of sentences in the content. | [46] |

Table 13: Quality signals based on ML heuristics.

| Annotation Tag | Description | Reference(s) |
|---|---|---|
| rps_doc_books_importance | Given a bag of 1,2-wordgram model trained on Books $p$, and a model trained on the source domain $q$, This is the logarithm of the ratio $p/q$. | [62] |
| rps_doc_openwebtext_importance | Given a bag of 1,2-wordgram model trained on OpenWebText $p$, and a model trained on the source domain $q$, this is the logarithm of the ratio $p/q$. | [62] |
| rps_doc_wikipedia_importance | Given a bag of 1,2-wordgram model trained on Wikipedia articles $p$, and a model trained on the source domain $q$, this is the logarithm of the ratio $p/q$. | [62] |
| rps_doc_ml_wikiref_score | Fasttext classifier prediction for the document being a Wikipedia reference. This is the same fasttext model used in the RedPajama-1T dataset. Only applies to English data. | [57] |
| rps_doc_ml_palm_score | Fasttext classifier prediction for the document being a Wikipedia article, OpenWebText sample or a RedPajama-V1 book. Only for English data. | [12], [16] |
| rps_doc_ml_wikipedia_score | Fasttext classifier prediction for the document being a Wikipedia article. This is used for non-English data | - |

Table 14: Summary of Quality signals which measure how repetitive text is.

| Annotation Tag | Description | Reference(s) |
|---|---|---|
| rps_doc_frac_chars_dupe_10grams | The fraction of characters in duplicate word 10grams. | [43, 45] |
| rps_doc_frac_chars_dupe_5grams | The fraction of characters in duplicate word 5grams. | [43, 45] |
| rps_doc_frac_chars_dupe_6grams | The fraction of characters in duplicate word 6grams. | [43, 45] |
| rps_doc_frac_chars_dupe_7grams | The fraction of characters in duplicate word 7grams. | [43, 45] |
| rps_doc_frac_chars_dupe_8grams | The fraction of characters in duplicate word 8grams. | [43, 45] |
| rps_doc_frac_chars_dupe_9grams | The fraction of characters in duplicate word 9grams. | [43, 45] |
| rps_doc_frac_chars_top_2gram | The fraction of characters in the top word 2gram. | [43, 45] |
| rps_doc_frac_chars_top_3gram | The fraction of characters in the top word 3gram. | [43, 45] |
| rps_doc_frac_chars_top_4gram | The fraction of characters in the top word 4gram. | [43, 45] |

Table 15: Summary of Quality signals which are based on the content of the text, measuring toxicity.

| Annotation Tag | Description | Reference(s) |
|---|---|---|
| rps_doc_ldnoobw_words | The number of sequences of words that are contained in the List-of-Dirty-Naughty-Obscene-and-Otherwise-Bad-Words blocklist. The blocklist is obtained from https://github.com/LDNOOBW/List-of-Dirty-Naughty-Obscene-and-Otherwise-Bad-Words. | [46] |
| rps_doc_ut1_blacklist | A categorical id corresponding to the list of categories of the domain of the document. Categories are obtained from https://dsi.ut-capitole.fr/blacklists/ | [44] |

Table 16: Examples of documents and corresponding cluster topics from Nomic Atlas [41].

| Cluster Topics (broad - medium - specific) | Document |
|---|---|
| Election - Health (2) - COVID Testing | immediately moving to the Purple Tier. This is the most restrictive level in the State's effort to control the spread of COVID-19. Businesses and residents must comply with the Purple Tier restrictions by Tuesday, Nov. 17. To determine restrictions by industry, business and activity, visit: https://covid19.ca.gov/safer-economy/ Read the full news release here: www.gov.ca.gov/2020/11/16/governor-newsom-announces-new-immediate-actions-to-curb-covid-19-transmission/ Watch the Governor's press conference during which he made the announcement today here: www.facebook.com/CAgovernor/videos/376746553637721 According to County of Orange officials, schools that have not already opened must continue with remote classes and cannot reopen in-person. Read the County's release here: https://cms.ocgov.com/civicax/filebank/blobdload.aspx?BlobID=118441 The California Department of Public Health has also issued a travel advisory encouraging Californians to stay home or in their region and avoid non-esse |
| Religion/Spirituality - Gaming - Gaming (3) | Top 100 Employers, and one of Canada's Top Employers for Young People multiple years running! At Ubisoft Toronto, we look for people who are excited to create the future of games in one of the most diverse cities in the world. We believe that embracing our differences helps us build stronger creative teams and develop better games for all players. We are an equal-opportunity employer and welcome applications from all interested candidates. We strongly encourage applications from Indigenous people, racialized people, neurodivergent people, people with disabilities, people from gender and sexually diverse communities and/or people with intersectional identities. We are committed to providing reasonable accommodation for people with disability upon request. If this sounds like your kind of studio, what are you waiting for? Apply to join us now! We thank you for your interest, however, only those candidates selected for an interview will be contacted. No agencies please. Senior Game Design |
| Education - Golf - Rotary Meetings | what's happening. Conversely, some people rely on the newsletter. Thus, the more avenues to inform people, the better. attendance at many social functions is poor, possibly due to the limited advertising reach. In practical terms, it means that social functions may be advertised in the OOC newsletter (current practice) the schedule, as is done for outdoor activities such as hikes the OOC's Facebook group As when social functions are advertised in the newsletter, the person organizing the social function can choose how much location information to provide, especially if it is to be held at someone's residence. OOC bylaw Article 3, Section 9 (f) states (highlighting added) (f) Social Coordinator: Shall be responsible for coordinating all social events for Club members only, and for preparing a schedule of these outings, not to be advertised to non-members. The executive voted to amend this statement by removing the limitation per Paragraph 3 of "Article 5 - Amending Formula" of the Const |

Table 17: Examples of documents and corresponding cluster topics from Nomic Atlas [41].

| Cluster Topics (broad - medium - specific) | Document |
|---|---|
| Online Privacy - Privacy Policy - Contracts | shall be governed by the laws of the Federal Republic of Germany under exclusion of the UN Convention on the International Sale of Goods (CISG), without prejudice to any mandatory conflict of laws and consumer protection provisions. 11.2 If the Customer is an entrepreneur according to Sec. 14 German Civil Code ("BGB"), a legal person under public law or a special fund under public law the courts at the place of business of the vendor shall have exclusive jurisdiction in respect of all disputes arising out of or in connection with the relevant contract. 11.3 In the event that one or more provisions of the contract should be or become invalid or unenforceable, the validity of the remaining provisions shall not be affected thereby. The invalid or unenforceable provision shall be deemed to be replaced - as existent - with statutory provisions. In case of an unacceptable rigor to one of the parties, the contract shall be deemed invalid as a whole. 11.4 In case of deviations of these General |
| Religion/Spirituality - Film/Movie - Movie | Movie of Nelson Mandela's life premieres in South Africa Nov. 04 - Stars Idris Elba and Naomie Harris attend the premiere of "Mandela: Long Walk to Freedom," based on the autobiography of anti-apartheid icon Nelson Mandela. Matthew Stock reports. |
| Election - Election (2) - Healthcare (4) | McAuliffe revived that language as an amendment to the budget. He also called on the General Assembly to immediately convene a special joint committee that had been created to assess the impact that repealing the ACA would have had on Virginia. The legislature will gather April 5 to consider the governor's amendments and vetoes, but leaders said Monday that McAuliffe's new budget language stands no better chance this time. In a joint statement, the Republican leadership of the House of Delegates said expanding Medicaid would lead to increased costs and eventually blow a hole in the state budget. "The lack of action in Washington has not changed that and in fact, the uncertainty of federal health policy underscores the need to be cautious over the long term," the leaders, including House Speaker William J. Howell (R-Stafford) and the man selected to replace him as speaker when he retires next year, Del. Kirk Cox (R-Colonial Heights), said via email. "Virginians can barely afford our cu |

Table 18: Evaluations for the 468M parameter LM for different dataset filters and other strong web datasets. The top-scoring dataset for each metric is indicated in **_bolded underlined_**, the top-2 is **bolded**, and the third-scoring dataset is in _italics underlined_.

| Dataset | Deduplication | | Rule-based | | ML Heuristics | | | Natural Language Inference | | | Coref. Res. | Sentence Completion | |
|---|---|---|---|---|---|---|---|---|---|---|---|---|---|
| | Exact | Fuzzy | C4 | Gopher | Classif. | DSIR | PPL | ANLI | ARC-c | ARC-e | Winogrande | Hellaswag | LAMBADA |
| C4 | | | | | | | | 33.8 | 22.0 | 37.0 | 51.9 | **_32.9_** | 15.5 |
| Dolma-v1.7 CC | | | | | | | | 33.5 | **24.0** | 38.3 | 49.6 | 32.3 | 17.3 |
| FineWeb | | | | | | | | 34.0 | 23.4 | 37.7 | 51.8 | **32.8** | 18.1 |
| RefinedWeb | | | | | | | | 32.8 | 22.6 | 38.3 | 51.9 | 31.6 | 17.8 |
| RPv1-CC | | | | | ✔ (Wiki-Ref.) | | | 33.9 | 22.4 | 37.5 | **52.6** | 29.7 | _19.0_ |
| RPv2 (2023-14) | | | | | | | | 33.3 | 22.2 | 38.5 | 52.4 | 31.5 | 18.2 |
| RPv2 (2023-14) | ✔ | | | | | | | 33.9 | 22.1 | 38.1 | 50.6 | 31.3 | 18.0 |
| RPv2 (2023-14) | | ✔ | | ✔ (full) | | | | 34.1 | 22.3 | 38.3 | 52.2 | 32.1 | 18.7 |
| RPv2 (2023-14) | | ✔ | ✔ | | | | | 33.4 | 22.7 | 38.9 | 51.1 | 32.4 | 17.5 |
| RPv2 (2023-14) | | ✔ | | ✔ (natlang) | | | Wiki-middle | 33.4 | **_24.2_** | 37.7 | 49.8 | 33.1 | **19.2** |
| RPv2 (2023-14) | | ✔ | | ✔ (Rep.) | | | Wiki-middle | 34.2 | 23.1 | 37.4 | 50.8 | 32.5 | 18.5 |
| RPv2 (9 Dumps) | | ✔ | ✔ | | | | | _34.3_ | 23.5 | _38.6_ | 51.5 | 32.0 | 17.2 |
| RPv2 (9 Dumps) | | ✔ | ✔ | ✔ (full) | | | | 33.5 | 23.3 | 38.4 | 50.2 | **32.8** | 16.8 |
| RPv2 (9 Dumps) | | ✔ | ✔ | ✔ (Rep.) | | ✔ (Palm-mix) | | 33.8 | 21.9 | 38.0 | _52.5_ | 32.0 | 17.3 |
| RPv2 (9 Dumps) | | ✔ | ✔ | ✔ (Rep.) | ✔ (Palm-mix) | | | **34.6** | 23.3 | _38.6_ | 52.2 | _32.7_ | 16.4 |
| RPv2 (9 Dumps) | | ✔ | ✔ | ✔ (natlang) | ✔ (Palm-mix) | | | **_34.8_** | 23.0 | **_39.2_** | **_53.0_** | 32.3 | 16.9 |
| RPv2 (9 Dumps) | | ✔ | ✔ (line-filter) | ✔ (natlang) | ✔ (Palm-mix) | | | 33.7 | 22.9 | 38.5 | 50.9 | 32.3 | **_19.9_** |
| RPv2 (9 Dumps) | | ✔ | custom-rules | | ✔ (Wiki-Ref.) | | $P_{wiki} > 30$ | 33.2 | 23.0 | 37.9 | 49.6 | 30.1 | 18.7 |
| RPv2 (9 Dumps) | | ✔ | custom-rules + Gopher-Rep | | ✔ (Wiki-Ref.) | | $P_{wiki} > 30$ | 33.0 | _23.8_ | **38.9** | 50.5 | 30.0 | 18.9 |

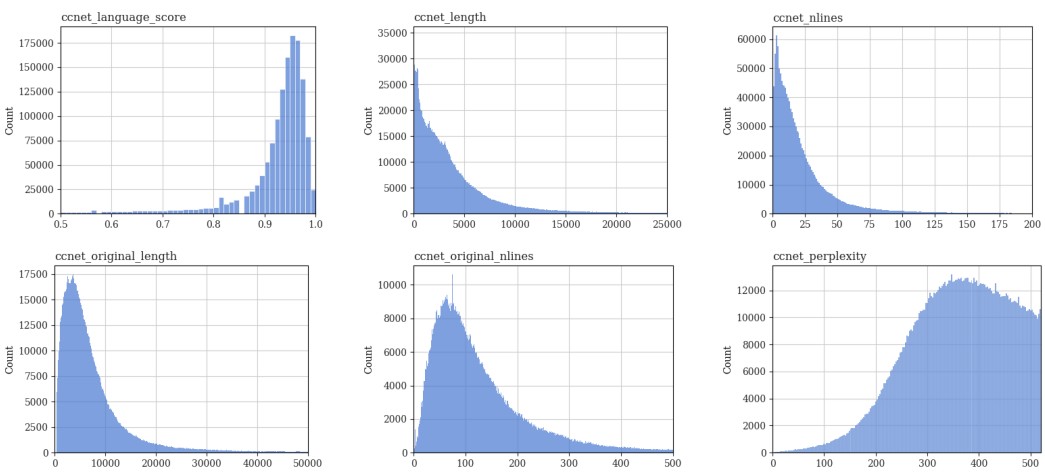

Figure 4: Histograms for the quality signals computed by the CCNet [61] pipeline.

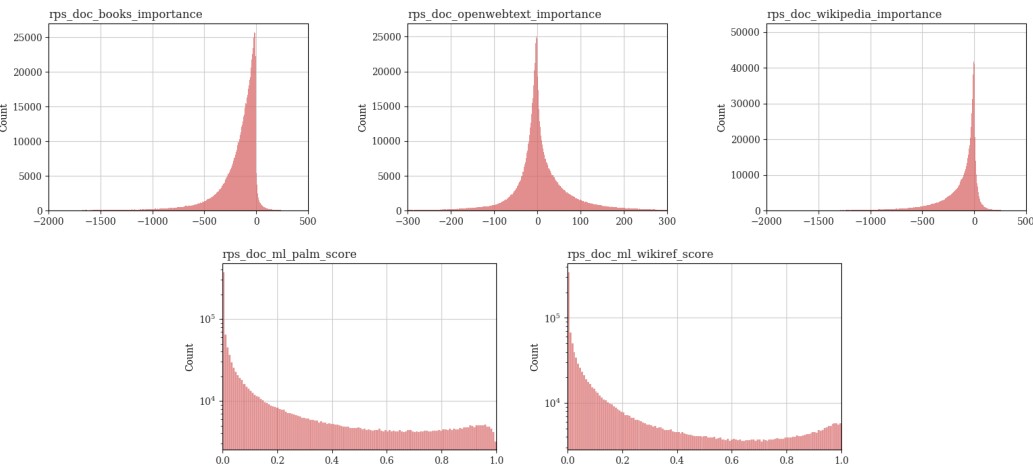

Figure 5: Histograms for ML-based quality signals.

Table 19: Evaluations in the 5-shot setting on MMLU and subtasks for the 468M parameter LM. The top-scoring dataset for each metric is indicated in **bolded underlined**, the top-2 is **bolded**, and the third-scoring dataset is in *italics underlined*.

| Dataset | Deduplication | | Rule-based | | ML Heuristics | | | MMLU | Stem | Humanities | Other | Social Sciences |
|---|---|---|---|---|---|---|---|---|---|---|---|---|
| | Exact | Fuzzy | C4 | Gopher | Classif. | DSIR | PPL | | | | | |
| C4 | | | | | | | | 24.9 | 26.4 | 24.1 | 25.8 | 23.4 |
| Dolma-v1.7 CC | | | | | | | | 26.0 | 27.8 | 24.5 | 26.2 | 26.1 |
| FineWeb | | | | | | | | 26.2 | 25.4 | 25.1 | 25.8 | *29.3* |
| RefinedWeb | | | | | | | | 24.8 | 23.9 | 23.7 | *26.5* | 25.6 |
| RPv1-CC | | | | | ✔ (Wiki-Ref.) | | | 25.1 | 25.1 | 23.7 | 24.0 | 28.5 |
| RPv2 (2023-14) | ✔ | | | | | | | *26.3* | 26.7 | **25.3** | 24.1 | **29.6** |
| RPv2 (2023-14) | ✔ | | | | | | | **26.4** | 26.8 | **25.3** | 25.2 | 28.8 |
| RPv2 (2023-14) | | ✔ | | ✔ (full) | | | | **27.0** | **28.8** | 24.8 | 25.6 | **30.0** |
| RPv2 (2023-14) | | ✔ | ✔ | | | | | 25.4 | 27.8 | 24.1 | 26.1 | 24.1 |
| RPv2 (2023-14) | | ✔ | | ✔ (natlang) | | | Wiki-middle | 26.1 | 27.4 | **25.2** | 24.6 | 27.7 |
| RPv2 (2023-14) | | ✔ | | ✔ (Rep.) | | | Wiki-middle | 25.5 | 24.3 | **25.2** | **27.8** | 24.8 |
| RPv2 (9 Dumps) | | ✔ | ✔ | | | | | *26.3* | **28.3** | **25.3** | 25.8 | 26.6 |
| RPv2 (9 Dumps) | | ✔ | ✔ | ✔ (full) | | | | 25.6 | *28.0* | *25.1* | 24.9 | 24.4 |
| RPv2 (9 Dumps) | | ✔ | ✔ | ✔ (Rep.) | | ✔ (Palm-mix) | | 24.4 | 26.9 | 23.7 | 24.8 | 22.7 |
| RPv2 (9 Dumps) | | ✔ | ✔ | ✔ (Rep.) | ✔ (Palm-mix) | | | 24.9 | 26.1 | 24.0 | 26.3 | 23.8 |
| RPv2 (9 Dumps) | | ✔ | ✔ | ✔ (natlang) | ✔ (Palm-mix) | | | 25.3 | 27.8 | 24.2 | 25.4 | 24.5 |
| RPv2 (9 Dumps) | | ✔ | ✔ (line-filter) | ✔ (natlang) | ✔ (Palm-mix) | | | 25.1 | 27.5 | 24.0 | 25.0 | 24.4 |
| RPv2 (9 Dumps) | | ✔ | custom-rules | | ✔ (Wiki-Ref.) | | $P_{wiki} > 30$ | **27.0** | 27.9 | *25.1* | 26.0 | **30.0** |
| RPv2 (9 Dumps) | | ✔ | custom-rules + Gopher-Rep | | ✔ (Wiki-Ref.) | | $P_{wiki} > 30$ | 25.9 | 25.8 | 24.3 | **27.1** | 27.2 |

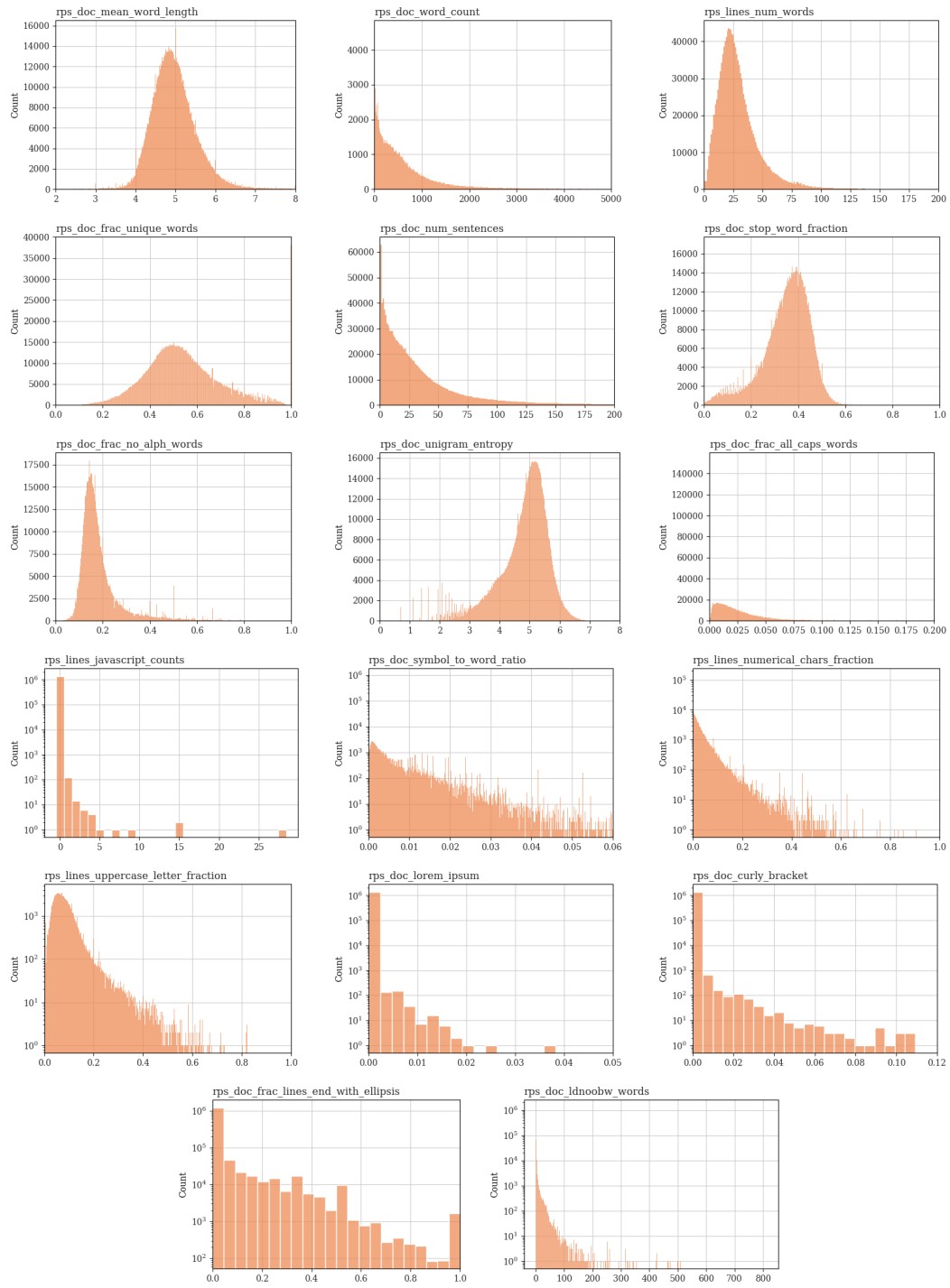

Figure 6: Histograms for Natural language based quality signals.

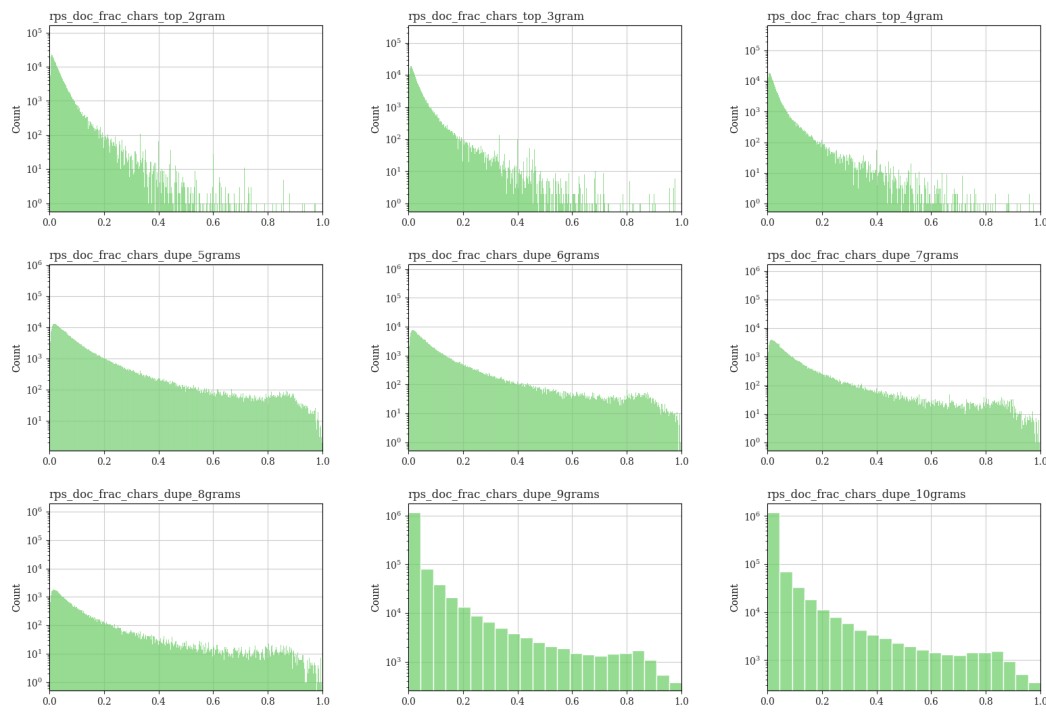

Figure 7: Histograms for quality signals measuring the repetitiveness of text.

Table 20: Evaluations on multiple choice tasks for the 468M parameter LM. The top-scoring dataset for each metric is indicated in **bolded underlined**, the top-2 is **bolded**, and the third-scoring dataset is in *italics underlined*.

| Dataset | Deduplication | | Rule-based | | ML Heuristics | | | CoQA | OpenbookQA | PIQA | PubMedQA | SciQ | SocialIQA | TruthfulQA |
|---|---|---|---|---|---|---|---|---|---|---|---|---|---|---|
| | Exact | Fuzzy | C4 | Gopher | Classif. | DSIR | PPL | | | | | | | |
| C4 | | | | | | | | 3.8 | **30.2** | *64.4* | 46.0 | 51.7 | *33.4* | 33.3 |
| Dolma-v1.7 CC | | | | | | | | 5.2 | 28.2 | **65.3** | 42.6 | 55.2 | 31.6 | 33.2 |
| FineWeb | | | | | | | | 9.0 | **29.4** | 64.5 | 41.4 | 54.3 | 32.4 | 33.5 |
| RefinedWeb | | | | | | | | **13.2** | 28.6 | *64.4* | *52.2* | **56.4** | 32.8 | 33.3 |
| RPv1-CC | | | | | ✔ (Wiki-Ref.) | | | 11.6 | 25.4 | 57.3 | 40.6 | **56.7** | 33.1 | **33.9** |
| RPv2 (2023-14) | ✔ | | | | | | | **12.5** | *29.2* | 61.6 | 40.8 | 53.0 | 32.9 | 31.4 |
| RPv2 (2023-14) | | ✔ | | | | | | 11.8 | 27.6 | 61.1 | 43.6 | 53.7 | 32.5 | 33.4 |
| RPv2 (2023-14) | | ✔ | ✔ | ✔ (full) | | | | 11.3 | 28.8 | 62.8 | 51.0 | 53.9 | 32.6 | 32.6 |
| RPv2 (2023-14) | | ✔ | ✔ | | | | | 5.8 | 28.8 | 63.4 | 49.6 | 54.7 | 36.6 | *33.8* |
| RPv2 (2023-14) | | ✔ | ✔ | ✔ (natlang) | | | Wiki-middle | 11.3 | 28.4 | 63.5 | 49.6 | 53.6 | 32.8 | 33.4 |
| RPv2 (2023-14) | | ✔ | ✔ | ✔ (Rep.) | | | Wiki-middle | *11.9* | **29.4** | 63.1 | **52.6** | 53.4 | 32.5 | 31.6 |
| RPv2 (9 Dumps) | | ✔ | ✔ | | | | | 6.6 | 29.0 | 62.0 | 36.2 | 53.7 | 33.2 | **34.3** |
| RPv2 (9 Dumps) | | ✔ | ✔ | ✔ (full) | | | | 5.8 | 28.6 | 62.8 | *51.2* | 54.8 | **34.4** | 31.2 |
| RPv2 (9 Dumps) | | ✔ | ✔ | ✔ (Rep.) | | ✔ (Palm-mix) | | 6.0 | **29.4** | 61.6 | 45.4 | 52.2 | *33.4* | 33.1 |
| RPv2 (9 Dumps) | | ✔ | ✔ | ✔ (Rep.) | ✔ (Palm-mix) | | | 5.4 | **29.4** | 62.5 | 45.0 | 51.7 | **34.0** | 33.7 |
| RPv2 (9 Dumps) | | ✔ | ✔ | ✔ (natlang) | ✔ (Palm-mix) | | | 4.9 | 28.0 | 62.9 | **52.8** | 52.0 | 33.0 | 33.6 |
| RPv2 (9 Dumps) | | ✔ | ✔ (line-filter) | ✔ (natlang) | ✔ (Palm-mix) | | | 6.4 | 27.0 | 63.2 | 47.8 | 52.9 | 32.8 | 32.0 |
| RPv2 (9 Dumps) | | ✔ | custom-rules | | ✔ (Wiki-Ref.) | | $P_{wiki} > 30$ | 10.0 | 27.8 | 59.6 | 41.2 | *55.8* | 33.3 | 32.0 |
| RPv2 (9 Dumps) | | ✔ | custom-rules + Gopher-Rep | | ✔ (Wiki-Ref.) | | $P_{wiki} > 30$ | 9.3 | 28.0 | 59.2 | 43.4 | 54.9 | 33.0 | 33.3 |

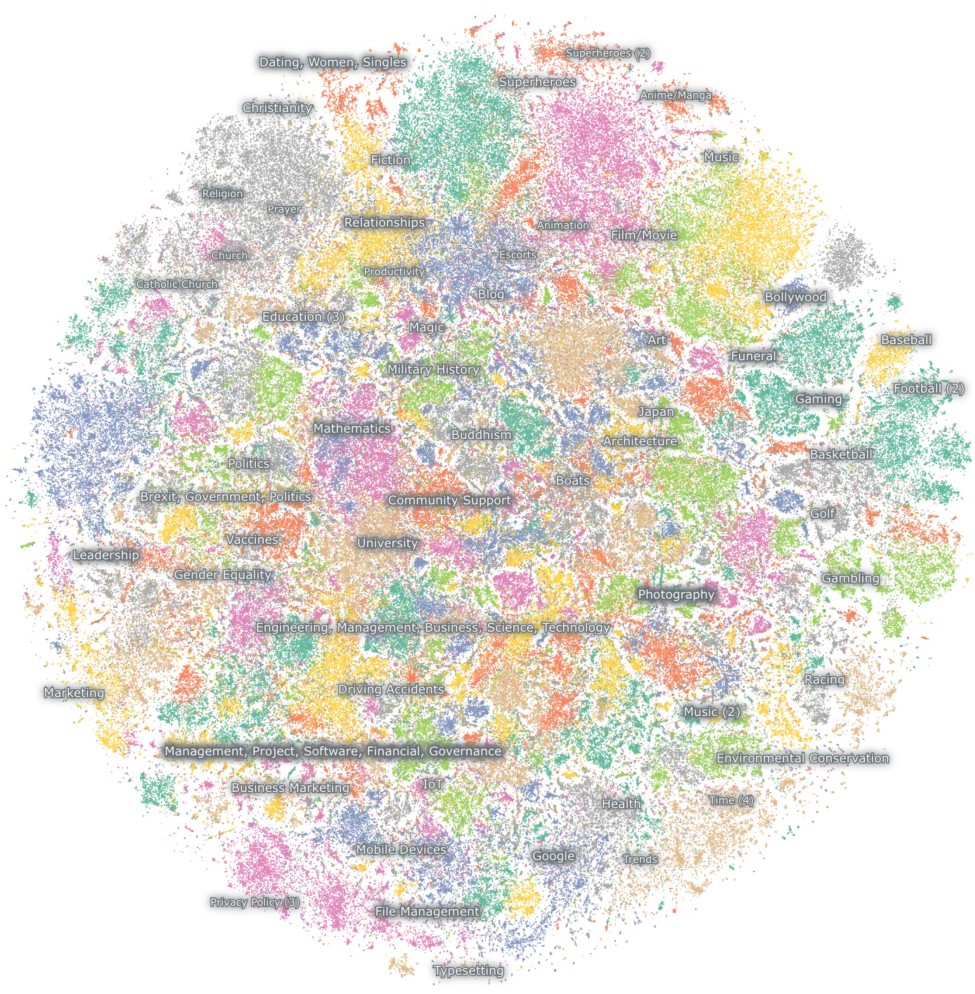

Figure 8: Visualization of topical clusters appearing in the RedPajama-V2 dataset. The clusters are computed in Nomic Atlas [41] based on gte-large-en-v1.5 embeddings for 2M documents of the unfiltered 2021-04 snapshot.

Table 21: Downstream task accuracy for a 1.6B LM trained on different datasets over 350B tokens.

| Dataset | Fuzzy Deduplication | Rule-based | | ML Heuristics | Natural Language Inference | | | Coref. Res. | Sentence Completion | |
| | | C4 | Gopher | | ANLI | ARC-c | ARC-e | Winogrande | Hellaswag | LAMBADA |
|---|---|---|---|---|---|---|---|---|---|---|
| RefinedWeb | | | | | 33.6 | 26.9 | 51.7 | 54.4 | 55.8 | 47.9 |
| RPv2 (full) | ✔ | | ✔ | WikiRef | 32.4 | 27.9 | 51.3 | 56.4 | 47.4 | 47.4 |
| RPv2 (full) | ✔ | ✔ | ✔(natlang) | Palm-Mix | 33.6 | 28.7 | 52.4 | 54.5 | 53.1 | 42.9 |

## E.5 Evaluations for the 1.6B Parameter Models

Tables 21, 22, and 23 show results for ablations with the 1.6B models. Each model was trained on 350B tokens.

Table 22: Evaluations in the 5-shot setting on MMLU and subtasks for the 1.6B parameter LM.

| Dataset | Fuzzy Deduplication | Rule-based C4 | Rule-based Gopher | ML Heuristics | MMLU MMLU | MMLU Stem | MMLU Humanities | MMLU Other | MMLU Social Sciences |
|---------|---------------------|---------------|-------------------|---------------|------|------|------------|-------|----------------|
| RefinedWeb | | | | | 25.3 | 24.9 | 24.9 | 27.0 | 24.7 |
| RPv2 (full) | ✔ | | ✔ | WikiRef | 25.2 | 26.0 | 26.7 | 23.9 | 23.3 |
| RPv2 (full) | ✔ | ✔ | ✔(natlang) | Palm-Mix | 24.7 | 25.7 | 25.4 | 23.8 | 23.4 |

Table 23: Evaluations on multiple choice tasks for the 1.6B parameter LM.

| Dataset | Fuzzy Deduplication | Rule-based C4 | Rule-based Gopher | ML Heuristics | CoQA | OpenbookQA | PIQA | PubMedQA | SciQ | SocialIQA | TruthfulQA |
|---------|---------------------|---------------|-------------------|---------------|------|------------|------|----------|------|-----------|------------|
| RefinedWeb | | | | | 47.4 | 31.6 | 73.8 | 57.0 | 75.3 | 41.0 | 36.6 |
| RPv2 (full) | ✔ | | ✔ | WikiRef | 43.7 | 32.6 | 67.4 | 55.6 | 72.7 | 40.4 | 36.9 |
| RPv2 (full) | ✔ | ✔ | ✔(natlang) | Palm-Mix | 22.1 | 32.2 | 71.3 | 55.2 | 71.0 | 42.2 | 35.7 |

# F Author Responsibility Statement

This aggregated dataset is licensed to you under the terms of the ODC-By-1.0, as well as any licenses that may apply to its constituent parts.

While we have made every effort to ensure the accuracy and legality of the data contained within this dataset, we cannot guarantee its absolute completeness or correctness due to its scale. Therefore, in the event that any rights, legal or otherwise, are violated through the use of this dataset, including but not limited to copyright infringement, privacy violations, or misuse of sensitive information, we, the authors, assume no liability for such violations. The dataset is provided to you "as is", without warranty of any kind, express or implied.

By utilizing this dataset, you agree that any consequences, legal or otherwise, arising from the use of this dataset will be your sole responsibility. You acknowledge that you will exercise due diligence and adhere to all applicable laws, regulations, and ethical guidelines when using the dataset. By accessing, downloading, or using this dataset, you signify your acceptance of this statement and your commitment to abide by the terms and conditions of the licenses.

# G License

The code provided in the GitHub repository [8] is distributed under an Apache 2.0 license. For the datasets themselves, we refer to the Common Crawl Foundation Terms of Use [9] for the datasets derived from the Common Crawl Archive. For the other datasets we refer to the license under which the dataset was originally distributed. Specifically,

- The C4 dataset at `https://huggingface.co/datasets/allenai/c4#license`,
- The GitHub subset was limited to MIT, BSD, or Apache licenses only,
- The Arxiv terms of use for the arxiv subset under `https://info.arxiv.org/help/api/tou.html`,
- The Wikipedia license for any wikipedia derived data `https://huggingface.co/datasets/legacy-datasets/wikipedia#licensing-information`,
- The StackExchange license on the Internet Archive for the StackExchange data `https://archive.org/details/stackexchange`.

We further request from users that they abide by each individual license for the subset they use.

---

[8] `https://github.com/togethercomputer/RedPajama-Data`
[9] `https://commoncrawl.org/terms-of-use`

