# OpenReview forum: "RedPajama: an Open Dataset for Training Large Language Models"
_NeurIPS.cc/2024/Datasets_and_Benchmarks_Track — NeurIPS 2024 Track Datasets and Benchmarks Spotlight_

### Official Review · Reviewer_zDq6 · 2024-06-17
**Great pretraining dataset**

**Rating:** 7
**Confidence:** 4
**Correctness:** Yes
**Clarity:** Yes

**Review:**

This paper presents the RedPajama(RP) dataset family with a model family training on top of RPv1, and a sweep of experiments testing different filter rules on RPv2. The delivered dataset has been proven valuable to the community.
My main concern to the paper is the lack of a dive-in study to the dataset, which makes the paper alike an engineering report about the output after applying a bunch of existing filter and deduplication methods from previous works (C4, Gopher, etc.)

**Strengths:**

1. The dataset family is proven valuable to the community, especially for the LLM pre-training.
2. The authors conduct a comprehensive analysis on different filter rules, making some novel findings.
3. The writing is clear to understand and easy to follow.

**Additional Feedback:**

N/A

**Documentation:**

This is well documented.

**Ethics:**

No ethical concerns.

**Limitations:**

Yes

**Opportunities For Improvement:**

1. a dive-in analysis on experiment results of Table.4 can help understand the dataset and filters better. For example, why does the benchmark score of a single dump outperforms 9 dumps? Why does single dump never applied C4 rules?
2. according to the supplementary material, the model size might be too small to compare the (filtered) dataset quality. For example, the MMLU score (table 16) is at most 27.0, which is too close to the random answer strategy (25.0). Besides, Table 17 suggests that FineWeb/RefineWeb outperforms RPv2 in many benchmarks.

**Relation To Prior Work:**

Yes

**Summary And Contributions:**

This paper presents RedPajama, a family of dataset for LLM pretraining. RedPajama 1.0 reproduced the data corpus of Llama 1.0, with a model family RedPajama-INCITE trained with all hyper-parameters from Llama. RedPajama 2.0 instead only focused on Web Data from CommonCrawl (CC), using multiple metrics to label web documents and performed experiments on a sweep of filter rules.

---

> ### Author Rebuttal · Authors · 2024-08-17
>
> We highly appreciate the thoughtful and thorough comments the reviewer has provided and believe that the concerns brought forward will help strengthen the contribution. The reviewer has expressed interest in seeing a more detailed analysis of the results of Table 4, as well as concerns around the scale of the ablation models.
>
> In response, we include (1) additional ablations with larger model sizes (1.6B parameters), and (2) a more thorough analysis of the results presented in Table 4. We have provided a detailed response to each point in the common rebuttal section since both were also brought up by other reviewers.
>
> ## Other Comments
>
> >For example, why does the benchmark score of a single dump outperforms 9 dumps? Why does single dump never applied C4 rules?
>
> These are interesting questions brought up by the reviewer. We believe that the difference in performance is mainly due to a different set of filters applied to the 9-dumps dataset, compared to the filters applied to the single dump. To test this hypothesis, we run another ablation where we apply the C4 rules to the 2023-14 dump so that (1) we include an additional row in Table 4 where C4 rules get applied to an individual dump, and (2) we see whether the performance difference stems from the source data (i.e., the dumps), or from the filters.

---

> > ### Author Rebuttal · Authors · 2024-09-01
> >
> > We would like to follow up on the answer provided above, as the additional experiment has now finished.
> >
> > Table 1 in the attached pdf will replace the current Table 4 in the paper. In this new table, we include an ablation where we have filtered the 2023-14 dump with C4 rules. From this, we can make the following additional conclusions:
> >
> > - it shows that the RPv2 2023-14 dump appears to perform slightly better in terms of aggregated downstream task accuracy than the RPv2 9 dumps dataset. This can be seen from comparing the two rows where the dataset was filtered with C4 rules and fuzzy deduplication.
> > - it additionally shows that the Gopher rules perform better than the C4 rules on the tasks considered. This can be seen from comparing the two rows corresponding to the 2023-14 dumps filtered with fuzzy deduplication and (1) Gopher rules vs. (2) C4 rules.
> >
> > We thank the reviewer for encouraging us to include this additional experiment.

---

### Official Review · Reviewer_CeY6 · 2024-07-19
**A Significant Contribution Toward Enhancing Transparency and Quality in Large Language Model Training Data**

**Rating:** 8
**Confidence:** 4
**Clarity:** Yes, the both paper and the supplemen…

**Review:**

The paper is well-written and represents a significant effort toward providing open access to high-quality benchmark training data for large language models. The data and code are accessible, and the authors have provided extensive explanations of the steps for downloading the data from HuggingFace or directly from their servers.
The prepared data comes with quality metrics, such as:
Natural Language Measures: Fraction of words in all caps, fraction of lines ending with terminal punctuation, and mean word length.
Repetitiveness: Fraction of characters in the most frequent n-grams and the fraction of characters in any duplicated n-grams.
Content-Based Filters: Number of sequences containing words from blocklists and flags indicating whether the domain of the document appears in known lists of blocked URLs.
I have a couple of minor comments that I recommend the authors consider in crafting the revised version:
Line 10: The abstract can be written in a clearer way. The current line suggests that V2 is a tag-along to V1, or that the manuscript was prepared for V1, and V2 was added at the last minute, which is not the case. I suggest mentioning that you are releasing V1, which is an open reproduction of the LLaMA training dataset, and V2, which is a massive web-only dataset.
Line 121: Please drop colloquial expressions and keep the manuscript formal.
Line 130: Why are you discarding the tail? I understand you are doing that because of poor perplexity scores, but the readers might not understand it. It would be good to add a sentence to explain this.
Line 131: What is CITE? Please fix the citation.
Line 139: Please add a citation for fastText. The citation is included in line 294, but it is more appropriate to include it at the first occurrence.
Line 140: Please clarify whether there is any report on the LLama dataset and its size. What do you mean by "approximately"? Do you know the exact numbers? If not, and how do you define "approximately"?
Line 315: You mentioned tail, middle, and head, but you did not explain their meanings. I suggest adding brief explanations to that section.
Since you mentioned the number of tokens (which is about 100 trillion), please also include the tokenizer's specific mechanism and vocabulary size that directly influence this token count. I assume you are using mistral-7B tokenizer (based on github issues and discussions), but it would be good to mention it in the paper.

**Strengths:**

The manuscript provides comprehensive documentation on the creation and curation processes of the datasets. This level of transparency is crucial for reproducibility and trust in the research community.

Both RedPajama-V1 and RedPajama-V2 datasets are openly accessible. This democratizes access to high-quality training data, enabling a more comprehensive range of researchers and organizations to utilize and build upon these resources.

The datasets include rigorous quality signals and filtering criteria, ensuring the inclusion of high-quality text data. This enhances the reliability and performance of models trained on these datasets.

RedPajama-V1 includes a diverse mix of data sources, such as CommonCrawl, GitHub, Wikipedia, and ArXiv, providing a rich and varied dataset for training. RedPajama-V2 offers a massive web-only dataset with comprehensive quality signals, facilitating the creation of customized, high-quality subsets.

The RedPajama-V2 dataset, with over 100 trillion tokens, is one of the largest publicly available datasets, supporting the training of large-scale language models.

The paper includes detailed evaluations of the RedPajama-INCITE family of models, highlighting their competitive performance. This provides clear evidence of the datasets' utility in training high-performing language models.

The datasets come with various quality metrics, such as natural language measures, repetitiveness, and content-based filters. These metrics help assess and ensure the quality of the data.

RedPajama-V2 includes text data across multiple languages, broadening the applicability and utility of the dataset for multilingual model training.

The authors provide extensive explanations and documentation, including detailed instructions for downloading and using the datasets. This enhances usability and encourages broader adoption.

By addressing the challenges of transparency, accessibility, and data quality, this work significantly contributes to the field of large language models, fostering further research and development.

**Additional Feedback:**

N/A

**Correctness:**

The claims made in the submission are correct and well-supported by the documentation provided. The datasets are constructed in a sound and robust manner, with comprehensive quality metrics and filtering criteria ensuring high-quality data. The evaluation methods and experiment design for the RedPajama-INCITE models are appropriate and performed correctly, demonstrating the effectiveness of the datasets.

**Documentation:**

The submission provides sufficient detail on data collection and organization, availability and maintenance, and ethical and responsible use. The datasets are well-documented, accessible, and maintained, with clear guidelines for ethical use. The benchmark evaluation methods are detailed, supporting reproducibility. To further enhance the submission, the authors could expand on ethical guidelines and explicitly compare their work to previous contributions in the main text. Overall, the submission meets the necessary criteria for detailed dataset and benchmark documentation and reproducibility.

**Ethics:**

No.

While the authors have made commendable efforts in addressing some ethical concerns, there are areas where further discussion and review are warranted. By providing more detailed information on data privacy, copyright compliance, data representativeness, bias mitigation, safety and security measures, environmental impact, and human rights considerations, the authors can ensure that their work adheres to ethical standards and guidelines.

**Limitations:**

The authors did their best to address the limitations and potential negative societal impacts of their work. They have provided comprehensive documentation and have discussed the challenges related to data transparency, quality, and accessibility. They have also mentioned the possible societal impacts of using large language models trained on their datasets.
While the authors have touched upon the societal impacts, a more detailed discussion on ethical considerations could enhance the paper. This includes potential misuse of the models, biases in the training data, and measures to mitigate these issues.
The limitations section could benefit from clearer explanations of specific challenges faced during the dataset creation and curation processes. This might include more details on the constraints of using CommonCrawl data, issues related to data deduplication, and the challenges of ensuring data quality across multiple languages.
Providing guidelines for the ethical use of the datasets and models would be valuable. This could include recommendations for avoiding biases, ensuring fair use, and respecting privacy and copyright laws.
Including a section on potential risks associated with the deployment of models trained on these datasets would be helpful. This could cover scenarios where the models might generate harmful content or reinforce societal biases and suggestions for how to monitor and mitigate these risks.
Encouraging community engagement to gather feedback on ethical practices and potential improvements in data handling could be beneficial.

**Opportunities For Improvement:**

The project includes both code and technical documents. While reproducing all results during the review process is not feasible, I suggest that the authors be proactive on the project GitHub repository by addressing open pull requests and resolving raised issues. Although I do not want to hold these submitted issues against this submission, I strongly recommend that the authors consider them to improve the quality of the data and documentation. Additionally, please add contribution guidelines to the Github repository for the open-source community.

**Relation To Prior Work:**

Yes. The paper effectively highlights the unique aspects and improvements of the RedPajama datasets compared to previous contributions. The emphasis on transparency, quality control, scale, and accessibility clearly differentiates this work and provides valuable advancements in the field of large language models.

**Summary And Contributions:**

The paper introduces the RedPajama datasets, which are significant contributions to the field of large language models (LLMs) by addressing key challenges such as transparency, access to high-quality data, and the availability of artifacts and metadata. There are two major datasets included in this benchmark: RedPajama-V1 and RedPajama-V2.
RedPajama-V1 is an open reproduction of the LLaMA training dataset. It includes data from multiple sources like CommonCrawl, GitHub, Wikipedia, and ArXiv. The authors detail the selection and filtering criteria, resulting in a dataset comprising approximately 1.2 trillion tokens.
RedPajama-V2 is a massive web-only dataset, consisting of over 100 trillion tokens of text data across multiple languages. It incorporates various quality signals and metadata to enable flexible and high-quality dataset creation. These quality signals help in filtering and curating the data to ensure it meets high standards.
The RedPajama project aims to democratize access to high-quality training data by providing open datasets with detailed documentation on their creation and curation processes. This approach helps to overcome the common issue of lack of transparency in the composition and curation strategy of pretraining data for LLMs.
The paper also discusses the RedPajama-INCITE family, which includes pretrained and instruction-tuned models at the 3B and 7B scales. These models were trained on the Summit supercomputer. The authors highlight the challenges encountered, such as the need for FP16 training due to hardware limitations, and present evaluation results showing these models' competitive performance compared to other open models.

---

> ### Author Rebuttal · Authors · 2024-08-17
>
> We appreciate the concerns brought to our attention by the reviewer, and are especially grateful for encouraging us to provide more context around the ethical use and societal impacts of the datasets and models. Here, we lay out our responses to these concerns and the corresponding additions we plan to make to the revised manuscript.
>
> ## Societal Impact and Guidelines for Ethical Uses of Datasets and Models
> In the revision, we include an additional section emphasizing the ethical use of the datasets and models, as well as the societal impact. Specifically, we include the following section:
>
> > ### Societal Impact and Guidelines for Ethical Use
> *The datasets presented in this work are intended to train large language models. With the release of the RedPajama datasets, we hope to provide both researchers and practitioners with large-scale, high-quality datasets, fostering both research in and applications of LLMs. We believe that an open approach to dataset release and curation, detailing individual processing steps, design decisions, data sources, weaknesses, and strengths of the datasets, is crucial for a democratic and fair development of the field. With the release of these datasets, we have relieved the community of significant costs related to dataset curation and development and thereby hope to foster a better understanding and open development of this technology.*
>
> > *When using the RedPajama datasets, it is crucial to adhere to ethical guidelines to mitigate potential harm. Before training, the dataset must undergo rigorous filtering to remove any personally identifiable information (PII) and content that may be biased, harmful, or inappropriate. This process should involve automated tools as well as human oversight to ensure thoroughness. Users of the dataset should also be mindful of the potential for unintended biases in the trained models and take steps to minimize their impact. Furthermore, models trained on RedPajama datasets, like any other language models, have limitations that should be taken into consideration. For example, these language models may not always provide accurate or relevant answers, particularly for questions that are complex, ambiguous, or outside of the range of its training data. We, therefore, recommend implementing appropriate safety mechanisms when deploying models trained on these datasets, which can, for example, be realized via alignment or in the form of moderator models like Llama Guard [3].*
>
> ## GitHub Repository
> We are in the process of addressing and closing outstanding GitHub issues and open pull requests. We will also add contribution guidelines to the public Github repository.
>
> ## Other Comments
> We thank the reviewer for bringing a number of further concerns to our attention.
>
> >Line 10: The abstract can be written in a clearer way. The current line suggests that V2 is a tag-along to V1, or that the manuscript was prepared for V1, and V2 was added at the last minute, which is not the case. I suggest mentioning that you are releasing V1, which is an open reproduction of the LLaMA training dataset, and V2, which is a massive web-only dataset.
>
> In the revised version, we replace the mentioned passage in the abstract with the following:
>
> > *“To address these challenges, we release RedPajama-V1, an open reproduction of the LLaMA training dataset. In an additional effort, we release a massive, web-only dataset, RedPajama-V2, which consists of raw, unfiltered text data but is enriched with quality signals and metadata to curate the raw dataset.”*
>
> >Line 130: Why are you discarding the tail? I understand you are doing that because of poor perplexity scores, but the readers might not understand it. It would be good to add a sentence to explain this.
>
> The reviewer is correct. We decided to discard the tail, because we found the text quality to be of poorer quality than the head and middle buckets. In the revision, we include the following explanation:
>
> >*“Here we only keep the “head" and “middle" buckets. We discard the “tail” bucket, consisting of documents with higher perplexity scores, as we found the text generally to be of poorer quality.”*
>
> >Line 140: Please clarify whether there is any report on the LLama dataset and its size. What do you mean by "approximately"? Do you know the exact numbers? If not, and how do you define "approximately"?
>
> Table 1 in the Llama paper [1] details the sampling proportions and number of epochs over each slice when training for 1.4T tokens. From this, it is possible to approximately calculate the token count in the dataset, which we found to be 1.26T tokens. With 1.2T tokens in RPv1, we did not exactly match the token count, hence the term “approximately.” We will make this clear in the revision and include the token count in both datasets.
>
> >Line 315: You mentioned tail, middle, and head, but you did not explain their meanings. I suggest adding brief explanations to that section. Since you mentioned the number of
> tokens (which is about 100 trillion), please also include the tokenizer's specific mechanism and vocabulary size that directly influence this token count. I assume you are using mistral-7B tokenizer (based on github issues and discussions), but it would be good to mention it in the paper.
>
> In the revised version, we will add an explanation of the meaning of the perplexity buckets in greater detail. Furthermore, we will include additional explanations on the vocabulary size (32k) and the mechanism used by the Mistral-7B BPE tokenizer [2].
>
> In the revision we will further fix the missing citations and remove comments and colloquial expressions.
>
> ## References
> [1] Touvron, Hugo, et al. "Llama: Open and efficient foundation language models." arXiv preprint arXiv:2302.13971 (2023).
>
> [2] Jiang, Albert Q., et al. "Mistral 7B." arXiv preprint arXiv:2310.06825 (2023).
>
> [3] Inan, Hakan, et al. "Llama guard: Llm-based input-output safeguard for human-ai conversations." arXiv preprint arXiv:2312.06674 (2023).

---

### Official Review · Reviewer_zjih · 2024-07-21
**Review of "RedPajama: an Open Dataset for Training Large Language Models"**

**Rating:** 6
**Confidence:** 3
**Correctness:** Yes
**Clarity:** Yes, but there are some suggested imp…

**Review:**

**Quality and Clarity**
The paper demonstrates good quality in its detailed descriptions of the dataset creation processes, which facilitates reproducibility and extensions. The authors clearly state their goals and design principles, providing a strong foundation for understanding their approach. However, the paper has some weaknesses in clarity, particularly in the organization of the related work section and minor writing issues including typos.

**Originality**
The work presents some novel contributions, particularly the inclusion of quality signals for data samples in RedPajama-V2. This feature allows for easy filtering and weighting of samples. However, the recreation of LLaMA's training dataset, while useful, is not highly novel.

**Significance of this work**
The RedPajama datasets, especially RedPajama-V2, represent a significant contribution to the field of LLM development. The large scale of the dataset (100+ trillion tokens) and the inclusion of quality signals make it a valuable resource for researchers and developers. The framework for ablation studies using quality filters is promising and with further experimentation and ablations could lead to important insights in dataset curation.

**Overall**
In conclusion, while the paper has some scope for improvement in organization and depth of analysis, it presents a significant contribution to the field through its large-scale, quality-annotated dataset. The work's primary strength lies in its potential to advance open-source LLM development by providing a versatile and transparent dataset resource.

**Strengths:**

1. The dataset's large scale, with over 30 trillion tokens, provides a substantial resource for training large language models. Its efficacy and impact on the research community is evidenced by its usage in training strong models like Salesforce XGen and AI2's OLMo.
2. The paper offers detailed descriptions of the dataset creation process, enhancing reproducibility and allowing for extensions. The quality signals provided in the dataset offer a simple method (eg: dataset filtering/sample weighting) to create newer data subsets for different use cases. Furthermore, the framework for ablation studies using the RedPajama-V2 dataset and quality signals is promising and with further experimentation could lead to important insights in dataset curation strategies.

**Additional Feedback:**

Some other questions which were unclear from reading the paper and could be explained more clearly:
- Line 189: "We also had to lower the learning rate compared to those reported in the LLaMa training" -- was this also for training stability?
- Line 218: "We hypothesize this was partly due to training with FP16, which does not allow us to use larger learning rates." -- could you provide the intuition behind this claim?
- Why is the dataset named RedPajama?

**Documentation:**

Yes

**Limitations:**

Yes

**Opportunities For Improvement:**

1. While "versatility" is one of the design principles and design decisions for the dataset incorporate this principle, the evaluation of experiments/studies using this dataset do not reflect it. It would be useful to add some metric for versatility in the proposed dataset and compare existing datasets along this dimension.
2. The related work section could be better organized to highlight limitations in previous datasets specifically with respect to the design principles discussed. A structured table summarizing prior work against relevant dimensions (eg: number of tokens) would enhance clarity and understanding.
3. The results indicate that training with the RedPajama dataset does not achieve LLaMA-level performance. To strengthen their claims, the authors should consider conducting additional experiments to test their hypotheses regarding performance gaps.
4. The results show that training with the RedPajama-v1 dataset still does not achieve LLaMa-level performance. While the authors offer potential hypotheses for these performance gaps, they do not conduct any experiments to test these hypotheses. Further experimentation could be done to demonstrate the utility of the RedPajama-v1 dataset.
5.  In describing the heuristic measures used in creating quality signals (eg: for detecting natural language), provide motivation/references for the proposed techniques.
6. The results and discussion for ablation studies with quality signals for RedPajama-v2 dataset is lacking. While they successfully show that using different signals leads to different results, they do not offer a thorough analysis of these observations. For example, some interesting questions to answer would be
    (i) Are there some quality signals which are always useful and could be universally recommended? That is, does inclusion of this quality filter improve performance on all metrics/datasets?
    (ii) Is there a relationship between quality signals and metrics? That is, does inclusion of a quality signal filter consistently improve metric1 and hurt metric2 across task types? Overall, I would recommend that the authors invest more in the ablation studies for the RedPajama-v2 dataset which would be useful to users looking to pretrain LLMs with the dataset.
7. Please address the typos in the paper. Eg: Page 4, line 121

**Relation To Prior Work:**

Yes, but there are some suggested improvements highlighted above

**Summary And Contributions:**

This paper proposes RedPajama, a family of datasets for large language model pretraining which address limitations in previous datasets (transparency, scale, versatility). RedPajama-v1 is an open reproduction of the training data used to train LLaMa. RedPajama-v2 is the largest open LLM pretraining dataset consisting of 100+ trillion tokens. These datasets have been used to train strong language models like Salesforce XGen and AI2's OLMo. By detailing their data pre-processing steps and including a diverse set of data sources and data quality signals, RedPajama datasets offer the capability to filter/weight the data for different downstream use cases.

---

> ### Author Rebuttal · Authors · 2024-08-17
>
> We greatly appreciate the thoughtful and thorough review the reviewer has provided. We believe that answering these concerns and including them in the revised manuscript will make our contribution even stronger. Below, we answer each concern separately and hope this addresses the reviewer’s concerns. We have provided comments on the concern about the depth of the analysis on ablation studies in the common rebuttal section.
>
> ## Versatility as a Guiding Principle
> One of the guiding principles of the RedPajama datasets is that they should be seen as modular in the sense that they are meant to be combined, curated, and further refined by the community in new ways. We refer to this principle as “versatility,” which motivates the following set of characteristics :
>
> - Inclusion of multiple domains of data (e.g., web, code, science) that can be mixed in different ways.
> - Publication of the raw, unfiltered web data in RPv2 so that the dataset can be further curated and filtered (e.g., specific topics can be extracted and refined).
> - Publication of a large number of quality signals for quality filtering.
>
> To better illustrate the design principles along the versatility dimension, we include a comprehensive table in the related work section, which compares RedPajama with other datasets along this axis and compares with other datasets (table 1 in the attched pdf).
>
> ## Organization of the Related Work Section
> We thank the reviewer for bringing to our attention that the related work section needs to be better organized. In the revised version of this paper, we will include a table in Section 2, providing a more accessible way of comparing to existing work. This is shown in table 1 in the attched pdf.
>
> ## Performance Mismatch between Llama and RedPajama-INCITE
> We believe that the performance mismatch between the RedPajama-INCITE model and the LLaMA 1 model is mainly due to the following two factors:
>
> **FP16 Precision.** Due to hardware (V100 GPUs), we had to use fp16 precision instead of bf16, which is more unstable [1] and requires a smaller learning rate than the one reported in the LLaMA paper (1.2e-4 vs. 3.0e-4). We believe that the use of a smaller learning rate has contributed to the lower performance. To test this hypothesis, in the revision, we plan to include an ablation where we train a 1B model in bf16 with a higher learning rate (5e-4), and one in fp16 with a lower learning rate (2.5e-4) and with loss scaling.
>
> **Dataset Curation.** The description of the dataset curation in [2] lacked details on the exact data curation strategy. We believe that RPv1 is not an exact reproduction of the training data used to train Llama 1. We believe that this has also contributed to the worse performance. In the revision, we provide a more thorough explanation of where uncertainties and design decisions in the data curation process arose so that potential mismatches are highlighted more prominently. In particular, we include a table to highlight the uncertainties and decisions we had to make during data preparation (table 2 in the attched pdf).
>
> ## Motivation and References for Quality Signals in RedPajama-V2
> The heuristic measures in RPv2 were selected by a thorough review of pretraining data literature and were consolidated into the list included in RPv2. In the revised version, we will include more details on each quality signal in the appendix and provide references and motivations in the main part of the paper.
>
> ## Additional Feedback
> > Line 189: "We also had to lower the learning rate compared to those reported in the LLaMa training" -- was this also for training stability?
>
> > Line 218: "We hypothesize this was partly due to training with FP16, which does not allow us to use larger learning rates." -- could you provide the intuition behind this claim?
>
> The reviewer is correct. The lower learning rate was required for training stability. Even though we applied loss scaling to prevent underflows in the gradients, fp16 precision still introduces more instability [1] during training. For this reason, we saw it necessary to use a smaller learning rate than the original llama training.
>
> ## Why is the dataset named RedPajama?
> The name was chosen after the book “Llama Llama Red Pajama” by Anna Dewdney – this provided us with a fitting name that also includes a hint at the relation this dataset has with the LLaMA models.
>
> ## References
> [1] Kalamkar, Dhiraj, et al. "A study of BFLOAT16 for deep learning training." arXiv preprint arXiv:1905.12322 (2019).
>
> [2] Touvron, Hugo, et al. "Llama: Open and efficient foundation language models." arXiv preprint arXiv:2302.13971 (2023).

---

### Official Review · Reviewer_TTPJ · 2024-07-23
**Review Comments**

**Rating:** 9
**Confidence:** 4
**Correctness:** Yes
**Clarity:** Yes

**Review:**

Pros:
- The paper focuses on an interesting and important problem of improving the accessibility of the pre-train data for LLMs. Plenty data details are well-illustrated by the authors.
- Aside from the well-curated data, the authors also designed abundant experiments to study what is important to high-quality data and what are the influences of data in different levels of quality on the models, which may inspire following works in pre-training.
- Interesting observations and conclusions are obtained from the empirical experiments.

Cons:
- No obvious major weaknesses.

**Strengths:**

- The paper is well-written and easy-to-follow. Good visualizations can help people have better understanding towards data curation and model performances.
- The paper is aiming to explore an important problem for NLP community: (1) what are the high-quality data for pretraining stage of LLMs, (2) how to obtain them from web data, and (3) how to monitor the quality of crawled data from open web.
- The paper conducted abundant experiments on different recepies of data for comparison and obtained a bunch of interesting observations and conclusions.
- Now, the RedPajama corpus has been widely used by both academia and industry, shedding lights to the following LLM developments in the future.

**Additional Feedback:**

N/A

**Documentation:**

The dataset and repository is well documented in the link: https://github.com/togethercomputer/RedPajama-Data?tab=readme-ov-file

**Limitations:**

Yes

**Opportunities For Improvement:**

- There seems to be some unremoved comments in the paper. I recommend the authors to fix this in the next version of manuscripts. For example:
  - Line 121: (how much shade do we want to throw at LLaMA?)
  - Line 131: (CITE)
- The scale of the model and data still seems to limited compared to the newly released models and pre-training data. And the filtering method seems to be hard to be scaled up as well.

**Relation To Prior Work:**

Yes

**Summary And Contributions:**

The paper presents the RedPajama dataset, an open dataset designed for pretraining large language models (LLMs). It aims to enhance the accessibility and quality of training data for researchers and developers in the field of natural language processing. The key contributions of the paper cover the following points:
- Dataset Creation: The paper details the process of creating the RedPajama-V1 dataset, which is positioned within the landscape of existing open pretraining datasets. It emphasizes the importance of transparency and reproducibility in dataset construction.

- Model Development: The paper also describes the training of the RedPajama-INCITE family of LLMs, which includes pretrained and instruction-tuned models. It discusses the technical challenges faced during training on the Summit supercomputer and how these were addressed.

- Quality Standards: RedPajama-V2 is introduced as a web-only dataset that aims to set new standards for high-quality web datasets, with a focus on providing robust filtering mechanisms and quality signals.

---

> ### Author Rebuttal · Authors · 2024-08-17
>
> We sincerely thank the reviewer for the thoughtful review and concerns brought to our attention. Below, we provide our answers to the concerns raised. We addressed the concern around the scale of ablation models in the global section of the rebuttal.
>
> ## Scalability of the Filtering Method
> Processing and filtering large amounts of web data (100TB+) is indeed a highly non-trivial undertaking and requires significant amounts of compute resources. This is also one of the core challenges RPv2 aims to address by providing a comprehensive collection (40+) of precomputed quality signals designed for filtering the dataset. This has the advantage of relieving the community from the burden of spending resources on the computation of these metrics so that subsets can be created by a mere filtering operation over the dataset which is orders of magnitude cheaper than the full preprocessing with CCNet and the computation of quality signals.
>
> ## Unremoved Comments & Typos
> We thank the reviewer for the thorough review and will remove the comments (L. 121)  and fix the missing citations (L. 131) in the revised version.

---

### Author Rebuttal · Authors · 2024-08-17

We sincerely thank the reviewers for their valuable feedback. We are happy to observe that the reviewers appreciated the extensive description of our dataset curation process (**TTPJ**, **zjih**, **CeY6**), the scope of our ablation studies (**TTPJ**, **zjih**, **zDq6**), as well as the broad use of RedPajama in academia and industry which demonstrates its value (**TTPJ**, **zjih**, **CeY6**, **zDq6**).

Here, we comment on common comments brought up by reviewers. Answers for concerns from individual reviewers are provided in the corresponding reviewer sections.

## Scale of Ablation Models
Two reviewers (**TTPJ** and **zDq6**) have expressed concerns about the model scale used in the ablations (sec. 4.3, table 4. and tables 15-17 in the Appendix). We agree with the reviewers that the scale of the models in the dataset ablations is relatively small compared to other studies (e.g., [1,2]) and that the paper would benefit from larger scale ablations.

The decision to choose a smaller model scale is motivated by the desire to cover a broad range of configurations of quality signals to show the breadth of the dataset while keeping the amount of compute at a reasonable level. At the same time, we already see differences in downstream task performance on various benchmarks, allowing us to make statements about quality filtering based on these models. Further evidence for this has been shown in the recent DataComp paper [3], where the 1x Chinchilla-optimal setting at 400M parameters was highly correlated with the 7B 1x Chinchilla-optimal setting (Pearson’s r = 0.885). In addition, by showing that dataset ablation studies at this scale are feasible, we hope to provide further evidence that research in pretraining data curation can be conducted with smaller compute budgets available to a wider community.

However, we believe that the contribution of this work can be made stronger by including ablation models of larger scale and trained on a larger token budget. To that end, we are currently training three additional 1.6B ablation models on 350B tokens. The three ablation datasets are

1. RefinedWeb
2. RPv2 filtered with fuzzy deduplication, full C4 rules, Gopher-natlang rules and the fasttext-palm classifier.
3. RPv2 filtered with fuzzy deduplication, full Gopher rules and the fasttext wiki-ref classifier.

We will include a discussion on the scale of the ablation models and the new results in the revised paper.

## More detailed Analysis on Ablations
Reviewers **zjih** and **zDq6** encouraged us to provide a more detailed analysis on the results from the ablation studies (Table 4). We agree with the reviewers that the analysis in the current manuscript is not thorough enough and thank them for encouraging us to improve this part of the paper. In the revised version, we add the following paragraph, which includes several interesting findings and best practices for curating datasets based on the quality signals:

> *"From Table 4, we can draw a series of conclusions on filtering the RedPajama-V2 dataset. First, we can see that the Gopher rules generally improve performance. In particular, we see that fuzzy deduplication and filtering with Gopher has the highest aggregated scores across all RPv2 datasets. In addition, both the average and normalized average benchmark scores are only second to RefinedWeb, while the rank-score is substantially higher than for RefinedWeb. Upon closer inspection, from Tables 15, 16, and 17, we can see that this dataset is always in the upper middle (minimum rank score 9 of 18), while RefinedWeb is performing worse on Hellaswag, ARC-e, LAMBADA, Winogrande, MMLU, and OpenBookQA. This indicates that filtering RPv2 with the full Gopher rules and fuzzy deduplication (Minhash LSH) creates a dataset that performs well across a wider range of tasks than all other datasets. Second, we can see that the Gopher-natlang filters perform better than the Gopher-repetition filters. Third, in the context of model-based filtering, we see no significant difference between using a fasttext classifier and DSIR. Fourth, using only the line-level C4 filters reduces Perplexity but has negligible effect on the (aggregated) benchmark scores. Fifth, including the C4 rules seems to degrade aggregated performance slightly. Finally, we notice that the unfiltered RPv2 2023-14 dataset appears to have the lowest perplexity on the Paloma dataset, while other filtering methods lead to models with higher perplexity. We believe that this can be (at least in part) attributed to the wide range of domains covered by Paloma. In addition, Paloma also contains the RPv1 dataset, which can explain the low Perplexity score obtained by the model trained on RPv1-CC. In conclusion, this series of ablation studies shows how the quality signals in the RPv2 dataset can be used to successively create better datasets, leading to high-quality web datasets for pretraining."*

Furthermore, we have revised the benchmark tasks that go into the aggregated scores (table 4) by excluding tasks for which all models are within less than 2% of the random baseline. The aggregated scores now take into account ARC-e, Winogrande, Hellaswag, LAMBADA, CoQA, MMLU, OpebookQA, PIQA, PubMedQA, and SciQ. In addition, next to the raw average score, we follow [3] and also include a normalized average score which is obtained by subtracting the random baseline from the achieved score and dividing it by the difference between the max score and the random baseline.

## References
[1] Penedo, Guilherme, et al. "The FineWeb Datasets: Decanting the Web for the Finest Text Data at Scale." arXiv preprint arXiv:2406.17557 (2024).

[2] Soldaini, Luca, et al. "Dolma: An open corpus of three trillion tokens for language model pretraining research." arXiv preprint arXiv:2402.00159 (2024).

[3] Li, Jeffrey, et al. "DataComp-LM: In search of the next generation of training sets for language models." arXiv preprint arXiv:2406.11794 (2024).

---

> ### Author Rebuttal · Authors · 2024-09-01
>
> We would like to follow up on the initial answer in the rebuttal since by now, the majority of additional experiments have concluded.
>
> In the attached pdf, we include the following set of tables with additional results:
>
> - Table 1 in the pdf will replace the current Table 4 in the paper. It contains an additional experiment where we filtered the 2023-14 dump with C4 rules, as requested by reviewer zDq6. This additional experiment shows that the RPv2 2023-14 dump appears to perform slightly better in terms of aggregated downstream task accuracy than the RPv2 9 dumps dataset. In addition, this experiment shows that the Gopher rules perform better than the C4 rules. We thank the reviewer for encouraging us to include this additional experiment as it allows us to draw more conclusions.
> - Tables 2-5 in the pdf contain results on larger models (1.6B parameters) trained for 350B tokens. From the table, we can see that the rankings between datasets are largely preserved, compared to the 500M parameter ablations: in both settings, RefinedWeb has the highest average benchmark accuracy, while the dataset filtered with Gopher continues to outperform the dataset filtered with a combination of C4 and a subset of Gopher rules.

---

### Decision · Program_Chairs · 2024-09-26

**Decision:**

Accept (Spotlight)

**Comment:**

The paper presents the RedPajama dataset that provides transparent and reproducible processes for pretraining LLMs. It integrates various quality signals that enable flexible dataset filtering and quality control, contributing significantly to the development of open LLMs. Extensive ablation studies were conducted in the work and they explore the impact of different filtering rules and quality signals on model performance, offering insights into dataset curation strategies. The reviewers had some concerns about the scale and depth of the ablation study experiments. The authors have put a lot of effort into addressing the concerns. The scale of the ablation study is the only concern that could not be addressed (understandably so due to the computational constraints). Overall, this is a very good contribution and meets the high NeurIPS benchmark track standard. All the reviewers have rated the paper highly. It is a great contribution to the research community. In fact, RedPajama datasets have already been widely adopted by academia and industry, supporting high-quality, large-scale LLM pretraining.